# FEW-SHOT CLASS-INCREMENTAL LEARNING BASED ON HIERARCHICAL DUAL-STREAM INTERACTION AND ASSOCIATIVE MEMORY FUSION

## ABSTRACT

Few-Shot Class-Incremental Learning (FSCIL) aims to learn novel classes from limited examples while preserving previously acquired knowledge. Current methods face two challenges: (1) Collapsed intra-class variance, where enhancing base-class separability limits generalization; and (2) Boundary instability, where few novel samples distort feature distribution and cause catastrophic forgetting. To address these challenges, we propose a cognition-inspired framework that employs a dual-stream network to extract a unified representation space with strong generalization and a hierarchical fusion mechanism with associative memory to improve old and new feature distribution. This framework comprises two key modules for rapid adaptation and long-term stability. The Hierarchical Dual-Stream Interaction Network (HDIN) decouples feature learning into a ResNet-based local stream for fine-grained detail extraction and a ViT-based global stream for long-range semantic dependencies. These streams are dynamically integrated via channel-adaptive attention to harmonize multi-scale information, simulating cognitive-level feature integration. The Associative-Enhanced Hierarchical Memory Fusion (AE-HMF) module simulates cortical memory consolidation by Gaussian sampling from class prototypes as associative memories and performing cross-layer feature interactions. Experiments on CIFAR100, miniImageNet, and CUB200 show that under the setting of no large-scale pretraining or data expansion techniques, our approach achieves the lowest Performance Decline Rates (DR) across all benchmarks, delivering a state-of-the-art balance between accuracy and forgetting. This work establishes a cognition-inspired, unified framework that effectively promotes the generalization capability and reduces catastrophic forgetting in FSCIL.

## 1 INTRODUCTION

Despite the success of deep neural networks in tasks such as image recognition and natural language processing, their reliance on large-scale, fully annotated data under a static, closed-world assumption limits their applicability in dynamic, data-scarce real-world settings (He et al., 2016a). In contrast, humans possess the ability to continually acquire new knowledge from limited examples and exhibit lifelong adaptability. It inspired the development of continual learning paradigms, such as Class-Incremental Learning (CIL), which enables models to integrate new classes without forgetting previously learned ones (De Lange et al., 2021; Zhou et al., 2024). To further approximate human-like adaptability, Few-Shot Class-Incremental Learning (FSCIL) requires models to incrementally learn from only a few examples per new class, thereby addressing more complex and dynamic real-world scenarios. However, FSCIL introduces an additional challenge due to the few-shot setting, which makes effective representation learning more difficult. This aggravated catastrophic forgetting due to decision boundary shifts and overfitting (Zhang et al., 2025).

In recent years, various methods have been proposed to address the FSCIL problem, typically leveraging pseudo class construction (Song et al., 2023; Chen et al., 2025), cognition-inspired representation learning (Wang et al., 2024b), or adaptive decision boundaries (Li et al., 2025). Despite these advances, they invariably encounter a critical trade-off: enhancing base-class discriminability often undermines transferability to new classes. This tension arises from two intrinsic issues. First,

conventional single-stream architectures encode mixed local details and global semantics within a unified feature space, which leads to collapsed intra-class variance and restricts generalization. Second, the paucity of novel-class examples not only distorts the overall feature distribution but also destabilizes established decision boundaries for base classes.

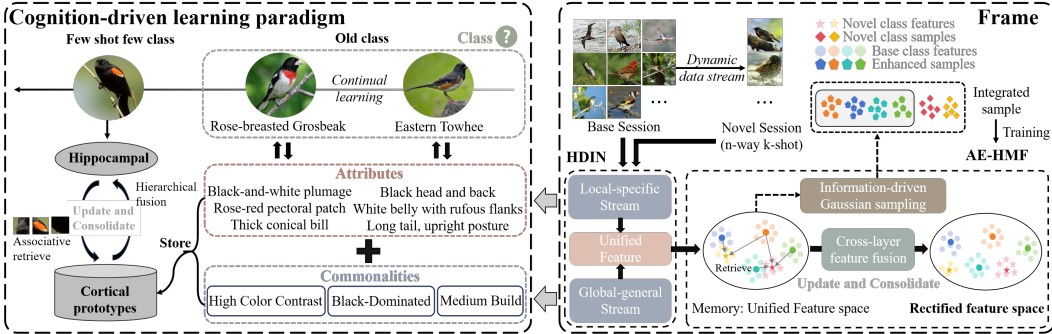

Figure 1: Cognition-inspired few-shot class-incremental learning framework. **Left:** human-like continual learning via integrating class-specific attributes and shared commonalities, and continuously updates and consolidates new knowledge by constructing associative memories through retrieval and cross-layer memory fusion. **Right:** HDIN integrates fine-grained details and global semantics through multi-granularity interaction, while AE-HMF fuses old and new knowledge with cross-layer feature fusion relying on retrieved Gaussian sampling to balance plasticity and stability.

Humans, by contrast, excel at rapidly assimilating new concepts from minimal examples while retaining existing knowledge. According to Feature Integration Theory, the visual system processes low-level feature attributes in parallel and subsequently binds them into coherent object representations via attentional mechanisms, with dynamic, bidirectional interactions between high and low-level processing stages (Treisman & Gelade, 1980; Goldstone et al., 1994). Moreover, once the brain forms a closed-loop encoding of event elements, even partial cues can trigger hippocampal activation and reconstruct complete experiences in the cortex (Horner et al., 2015). Research on temporal cognition demonstrates that the brain constructs memory for time by binding temporal sequences with event-based common structures, thereby supporting associative memory formation (Cohn-Sheehy & Ranganath, 2021). Predictive coding and replay mechanisms have also been proposed as means by which structured representations of common features facilitate learning and flexible reasoning (Momennejad, 2020). Hierarchical models of perception and memory suggest that local feature processing and global pattern recognition operate across multiple temporal and spatial scales in the brain (Hasson et al., 2015). Inspired by these neuroscientific insights, we designed the Hierarchical Dual-Stream Interaction Network (HDIN) to emulate the extraction of global and local features, and the Associative-Enhanced Hierarchical Memory Fusion (AE-HMF) mechanism to model associative memory formation (Figure 1).

HDIN, based on parallel and integrative visual processing, formulates a unified representation space through interactive encoding of complementary information granularity. It establishes a bidirectional interaction between a ResNet-based pathway for fine-grained discrimination and a ViT-based pathway for global semantic abstraction (He et al., 2016b; Dosovitskiy et al., 2020). The combination leverages the locality sensitivity of ResNet and the long-range dependency modeling capability of ViT, enabling richer and more balanced feature representations. These streams are dynamically fused via multi-granularity attention, allowing selective emphasis on attribute or commonality level cues. AE-HMF, motivated by cortical memory consolidation, integrates old and new knowledge through cross-layer feature interactions and information-guided Gaussian sampling from class prototypes as associative cues. By leveraging mutual information to select relevant prior knowledge, it models hippocampal-cortical loops that trigger memory reactivation. This alignment of multi-level features across sessions facilitates adaptive knowledge transfer and maintains semantic consistency, effectively balancing plasticity and stability under few-shot constraints. Extensive experiments demonstrate the effectiveness of our method, and visualization results confirm significant improvements in intra-class compactness and inter-class separability.

To summarize the contributions of this work:

- We introduce a cognition-inspired framework that models multilevel human learning by feature integration and associative memory to alleviate collapsed intra-class variance and boundary instability in FSCIL.

- We propose a dual-stream architecture that models bidirectional interactions between local fine-grained and global semantic features via multi-granularity attention, effectively disentangling local and global representations to alleviate feature collapse and enhance transferability.

- We introduce an associative memory fusion module that leverages cross-layer interaction and prototype-based Gaussian sampling as associative memory to align old and new knowledge, mitigating boundary drift and balancing stability and plasticity.

- Extensive experiments on *mini*ImageNet, CIFAR100, and CUB200 demonstrate that under the setting of no large-scale pretraining or data expansion techniques, our method achieves significant performance improvement with the lowest Performance Decline Rates (DR), outperforming recent state-of-the-art methods.

## 2 RELATED WORK

FSCIL aims to progressively expand the set of known classes using limited data while retaining knowledge acquired in earlier sessions. Existing methods can be categorized into data-based, structure-based, and optimization-based approaches.

Data-driven methods mitigate catastrophic forgetting through data replay or synthesis, including raw replay (Kukleva et al., 2021; Zhu et al., 2022), generative replay (Liu et al., 2022; Agarwal et al., 2022), and pseudo-scenario construction (Zhou et al., 2022a; Song et al., 2023; Chen et al., 2025). While FearNet (Kemker & Kanan, 2018) employs brain-inspired replay and FoCAL (Ayub & Fendley, 2022) utilizes Gaussian sampling to generate pseudo-samples, neither addresses strict few-shot incremental learning with cross-class covariance modeling. Distinct from these paradigms, our method avoids fine-tuning on replayed data and introduces structured covariance-aware fusion.

While FearNet [1] models memory replay in a brain-inspired manner and targets incremental class learning more broadly, it is not designed under the strict few-shot incremental constraints considered here. FoCAL [2] adopts a simpler per-class Gaussian sampling strategy and does not model cross-class covariance structure or feature disentanglement. In contrast, our approach does not fine-tune the model on replayed or generated samples.

Structure-based methods adapt model architectures or class relationships to preserve old knowledge and integrate new knowledge (Tao et al., 2020; Zhang et al., 2021; Yang et al., 2022). TOPIC (Tao et al., 2020) maintains a dynamic topological feature map via neural gas networks, and CEC (Zhang et al., 2021) models class prototypes as nodes in a graph attention network. However, such designs often incur increased computational overhead or require careful scheduling of structure updates.

Optimization-based methods focus on learning transferable and robust representations through refined training objectives (Zou et al., 2022; Yang et al., 2023; Zhao et al., 2023; Zhou et al., 2022b; Li et al., 2025; Oh et al., 2024). NCFSCIL (Yang et al., 2023) aligns features with predefined prototypes inspired by neural collapse, and CLOSER (Oh et al., 2024) encourages feature spread in a compact space. ADBS (Li et al., 2025) introduces adaptive class-specific decision boundaries and inter-class constraints. BiDistFSCIL (Zhao et al., 2023) uses dual teacher models for stability–plasticity balance, while LIMIT (Zhou et al., 2022b) employs a transformer-based meta-learning module for class relationship correction.

Our method falls within the category of structure-based approaches. Unlike prior work focusing on static graph design or class-wise topology, we focus on leveraging structured representation learning to bridge the distributional gap between base and novel classes. To this end, our method controls optimization of class relations and applies targeted alignment for novel classes by a transferable structural prior from base classes.

Compared with some existing methods, MCNet (Ji et al., 2023) integrates CNN and ViT at the model level, whereas our HDIN performs feature-level integration, enabling complementary representations through dual-stream interaction within a unified framework. CoDF (Wang et al., 2024b), inspired by cognitive principles, combines self-supervised learning with GMM-based structured

representations to enhance generalization. In contrast, our method introduces cross-layer feature interaction and Gaussian sampling to emulate memory consolidation, incorporating a memory fusion mechanism absent in CoDF. Yourself (Tang et al., 2024) proposed a new metric (gAcc) and utilized intermediate ViT features to improve the performance of new categories from the perspective of evaluation and feature reuse. Differently, our cognition-inspired framework alleviates intra-class variance collapse and boundary instability through a dual-stream interaction network and an associative-enhanced memory fusion module, improving separability and generalization.

## 3 METHODS

### 3.1 PROBLEM FORMULATION

In FSCIL, learning proceeds over sessions $\mathcal{D}^{(0)}, \mathcal{D}^{(1)}, \ldots, \mathcal{D}^{(T)}$, where each session $\mathcal{D}^{(t)} = (\mathbf{x}_i^{(t)}, y_i^{(t)}) \mid y_i^{(t)} \in \mathcal{C}^{(t)}$ introduces a disjoint class set. The base session ($t = 0$) provides abundant labelled data, while each incremental session ($t \geq 1$) adds $N$ novel classes with $K$ examples each ($N$-way $K$-shot). At time $t$, only $\mathcal{D}^{(t)}$ is available, and the model must retain accuracy over all seen classes, making stability–plasticity trade-offs challenging.

Following prior work (Zhang et al., 2021), we adopt a prototype-based classifier. In the base session, logits are computed via cosine similarity between normalized features $f_\theta(x)$ and class prototypes $\phi$:

$$z = T \cdot \left\langle \frac{f_\theta(x)}{|f_\theta(x)|_2}, \frac{\phi}{|\phi|_2} \right\rangle. \tag{1}$$

In incremental sessions, $f_\theta$ is frozen. For each novel class $i$, its prototype is the mean support feature:

$$p_i = \frac{1}{|D_i|} \sum_{x \in D_i} f_\theta(x), \tag{2}$$

where $D_i$ is the support set. These prototypes are appended to the classifier, and inference uses cosine similarity with the frozen extractor.

### 3.2 HIERARCHICAL DUAL-STREAM INTERACTION NETWORK

Human continual learning efficiently integrates both fine-grained local features and high-level global patterns. Inspired by this cognitive process, we propose the Hierarchical Dual-stream Interaction Network for FSCIL. As shown in Figure 2, HDIN consists of two complementary streams: the Local Specific Stream (LSS), which discriminative fine-grained information, and the Global Generic Stream (GGS), which extracts transferable semantic representations. These two branches interact through a multi-granularity fusion module, which constructs a unified representation space that enhances generalization while preserving existing knowledge.

The dual-stream architecture models the functional division observed in human vision, where a local pathway specializes in detecting fine-grained features, while a global pathway integrates broader contextual information. LSS adopts a ResNet-based architecture, leveraging residual connections to preserve local patterns such as textures and edges while ensuring stable gradient flow. GGS utilizes Vision Transformer blocks, which model global contextual relationships through self-attention mechanisms. By encoding local specificity and global generality, the model constructs a more robust and flexible representational space for FSCIL.

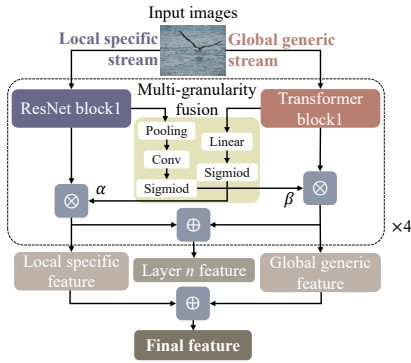

Figure 2: Illustration of the proposed Hierarchical Dual-stream Interaction Network (HDIN), which integrates local features from ResNet and global semantics from ViT via attention-based multi-granularity fusion. The Layer features are employed to support the operations of the AE-HMF module.

To align the cognitive representation of dual-stream features, we employ a multi-granularity fusion strategy based on SE attention, modeling channel dependencies to adaptively adjust the weights of local and global features for complementary integration. For local features $f_{\text{local}}$ and global features $f_{\text{global}}$, the channel attention weights are computed as follows:

$$
\begin{aligned}
\alpha &= \sigma(W_1 \cdot f_{\text{global}}), \\
\beta &= \sigma(W_2 \cdot \text{GAP}(f_{\text{local}})),
\end{aligned}
\tag{3}
$$

where $\alpha$ and $\beta$ modulate the channel importance of local and global features, respectively. $W_1$ is a fully connected layer, and $W_2$ is a $1 \times 1$ convolutional layer for channel transformation. $\sigma$ denotes the Sigmoid function, and GAP represents global average pooling. Spatial domain complementarity is achieved via cross-stream feature map convolution, yielding the fused feature representation:

$$
f_{\text{fuse}} = [\alpha \odot f_{\text{local}}, \beta \odot f_{\text{global}}].
\tag{4}
$$

### 3.3 Associative-Enhanced Hierarchical Memory Fusion

To improve the unstable boundary between the feature distribution of old and new classes, we propose the Associative-Enhanced Hierarchical Memory Fusion module, inspired by human cortical memory consolidation. As shown in Figure 3, AE-HMF dynamically integrates multi-level features through nonlinear associative correction, enhancing long-term memory retention and semantic stability. Inspired by the feedback modulation mechanism of the human visual cortex (Rao & Ballard,

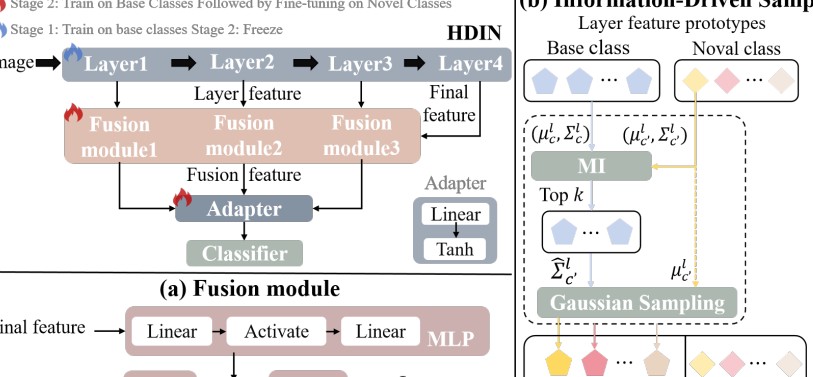

Figure 3: AE-HMF module working principle. During incremental learning, AE-HMF improves the highly generalized latent features extracted by HDIN through hierarchical cross-layer feature fusion blocks with associative memory. (a) The hierarchical cross-layer feature fusion block integrates hierarchical features with the final layer features. (b) Information-driven Gaussian sampling generates novel class enhanced samples, i.e., associative memory, by using the top-$k$ base class covariances selected via mutual information (MI), which are then used to train the cross-layer fusion modules.

1999), AE-HMF hierarchically fuses intermediate and final representations to correct latent feature distributions across multiple abstraction levels. Given intermediate-layer features $\mathbf{f}_{\text{inter}}^l \in \mathbb{R}^d$ and final-layer features $\mathbf{f}_{\text{final}} \in \mathbb{R}^d$, three MLP-based fusion modules perform transformation and fusion as follows:

$$
\begin{aligned}
\mathbf{f}_{\text{inter}}^{l'} &= \text{MLP}_{\text{inter}}(\mathbf{f}_{\text{inter}}^l), \\
\mathbf{f}_{\text{final}}' &= \text{MLP}_{\text{final}}(\mathbf{f}_{\text{final}}), \\
\mathbf{f}_{\text{fused}}^l &= \text{MLP}_{\text{fused}}([\mathbf{f}_{\text{inter}}^{l'}, \mathbf{f}_{\text{final}}']),
\end{aligned}
\tag{5}
$$

where $[\cdot, \cdot]$ denotes channel-wise concatenation. These fused vectors are further modulated by dynamically learned weights $\gamma^l$ from an adapter network. The final prediction is computed via normalized dot-product similarity over seen classes:

$$
\text{Pred} = \underset{\mathcal{D} \in \mathcal{D}^{(\leq t)}}{\arg\max} \left( \sum_{l=0}^{L} \gamma^l \left\langle \frac{\mathbf{f}_{\text{fused}}^l}{\|\mathbf{f}_{\text{fused}}^l\|_2}, \frac{\phi}{\|\phi\|_2} \right\rangle \right),
\tag{6}
$$

where $L$ is the number of considered layers. This formulation preserves cross-level context and complements the local-global synergy of LSS and GGS by aligning internal abstraction stages.

To ensure consistent representation learning across tasks, AE-HMF introduces a structured loss function that distinguishes between base and novel classes. For the base class, a cosine alignment loss is used to promote directional consistency between intermediate and final-layer features:

$$\mathcal{L}_{\text{cos}}^l = \frac{1}{N} \sum_{j=1}^{N} \begin{cases} 1 - \cos(\mathbf{f}_{\text{fused},j}^l, \mathbf{f}_{\text{final},j}), & y_j = 1 \\ \max(0, \cos(\mathbf{f}_{\text{fused},j}^l, \mathbf{f}_{\text{final},j})), & y_j = -1 \end{cases} \tag{7}$$

where $N$ is the number of training samples, $y_j \in \{1, -1\}$ indicates whether the alignment between the representations is encouraged (positive pair) or discouraged (negative pair), and $\cos(\cdot, \cdot)$ denotes cosine similarity. The total base loss is a weighted combination over all layers:

$$\mathcal{L}_{\text{base}} = \lambda_{\text{base}} \sum_{l=1}^{L} \gamma^l \mathcal{L}_{\text{cos}}^l, \tag{8}$$

where $\lambda_{\text{base}}$ is a scaling coefficient.

To enhance plasticity and mitigate overfitting in novel class learning, AE-HMF incorporates Gaussian sampling and two cross-entropy objectives. The similarity score between normalized feature $\mathbf{f}_{\text{fused},j}^l$ and corresponding class prototype $\phi_{cls}$ is:

$$z_{j,cls}^l = \left\langle \frac{\mathbf{f}_{\text{fused},j}^l}{\|\mathbf{f}_{\text{fused},j}^l\|_2}, \frac{\phi_{cls}}{\|\phi_{cls}\|_2} \right\rangle. \tag{9}$$

The losses for novel and global class predictions are:

$$\mathcal{L}_{\text{nov}}^l = -\frac{1}{N} \sum_{j=1}^{N} \sum_{i=1}^{K_{\text{nov}}} (y_{\text{nov}})_{j,i} \log \left( \frac{e^{z_{j,cls}^l}}{\sum_{k=1}^{K_{\text{nov}}} e^{z_{j,k}^l}} \right), \tag{10}$$

$$\mathcal{L}_{\text{glo}}^l = -\frac{1}{N} \sum_{j=1}^{N} \sum_{i=1}^{K_{\text{all}}} y_{j,i} \log \left( \frac{e^{z_{j,cls}^l}}{\sum_{k=1}^{K_{\text{all}}} e^{z_{j,k}^l}} \right), \tag{11}$$

where $y_{\text{nov}} \in \{0,1\}^{N \times K_{\text{nov}}}$ is the one-hot label for novel classes, and $y \in \{0,1\}^{N \times K_{\text{all}}}$ for all seen classes. $K_{\text{nov}}$ and $K_{\text{all}}$ are the number of novel and total classes, respectively. The total incremental learning loss is:

$$\mathcal{L}_{\text{inc}} = \sum_{l=1}^{L} \gamma^l \left( \lambda_{\text{cos}} \mathcal{L}_{\text{cos}}^l + \lambda_{\text{nov}} \mathcal{L}_{\text{nov}}^l + \lambda_{\text{glo}} \mathcal{L}_{\text{glo}}^l \right). \tag{12}$$

where $\lambda_{\text{cos}}$, $\lambda_{\text{nov}}$, and $\lambda_{\text{glo}}$ are hyperparameters balancing the importance of the corresponding loss terms.

### 3.4 INFORMATION-DRIVEN GAUSSIAN SAMPLING

To enhance the model's ability to generalize under FSCIL, we propose an information-driven Gaussian sampling mechanism based on multivariate Gaussian distributions, which leverages prior knowledge from base classes in a statistically grounded and information-theoretic manner.

Specifically, during the base training phase, we extract fused feature representations $\mathbf{f}_{\text{fused}}$ for each sample and compute the empirical mean $\boldsymbol{\mu}_c \in \mathbb{R}^d$ and covariance matrix $\boldsymbol{\Sigma}_c \in \mathbb{R}^{d \times d}$ for each base class $c \in \mathcal{C}_{\text{base}}$, which are fixed during the later phase. Each class can be modeled as a multivariate Gaussian distribution:

$$\mathcal{N}_c(\mathbf{x}) = \mathcal{N}(\boldsymbol{\mu}_c, \boldsymbol{\Sigma}_c). \tag{13}$$

During the incremental training phase, for each new class $c' \in \mathcal{C}_{\text{novel}}$, we extract its fused feature embeddings and compute the empirical distribution $\mathcal{N}_{c'}(\boldsymbol{\mu}_{c'}, \boldsymbol{\Sigma}_{c'})$. To guide Gaussian sampling,

we assess the statistical dependency between novel and base distributions using mutual information (MI), defined as:

$$I(X;Y) = H(X) + H(Y) - H(X,Y), \tag{14}$$

where $X \sim \mathcal{N}_{c'}$, $Y \sim \mathcal{N}_c$, and $H(\cdot)$ denotes the differential entropy of a multivariate Gaussian distribution with covariance $\boldsymbol{\Sigma} \in \mathbb{R}^{d \times d}$, given by:

$$H(X) = \frac{1}{2} \ln \left[ (2\pi e)^d |\boldsymbol{\Sigma}| \right]. \tag{15}$$

Based on the mutual information matrix, we select the top-$k$ base classes $\mathcal{C}_k$ with the highest scores, and leverage their covariance information to correct and update the distribution of the novel class, thereby generating new memory distributions that enrich the novel feature representation via associative inference. The covariance matrices of these selected base classes are averaged and regularized with a small diagonal term $\varepsilon I$ to ensure positive-definiteness. For each layer $l$, we construct the novel-class Gaussian distribution:

$$\widehat{\Sigma}_{c'}^{(l)} = \frac{1}{k} \sum_{b \in \mathcal{C}_k} \Sigma_b^{(l)} + \varepsilon I, \qquad \mathcal{N}_{c'}^{(l)} = \mathcal{N}\big(\mu_{c'}^{(l)}, \widehat{\Sigma}_{c'}^{(l)}\big). \tag{16}$$

Repeating the above statistical process over all feature layers independently yields a dictionary of per-layer multivariate normal generators (`MVN_distributions`), which serves as an associative memory bank providing distribution-consistent sampling for novel categories. The generated features are mixed with real few-shot samples and fed into the feature rectification and layer-attention modules during incremental training, enabling continuous adaptation while mitigating catastrophic forgetting. The pseudocode of the procedure of can be seen in Appendix B.

Under few-shot settings, the associative memory realized through Gaussian-based modeling not only offers stable auxiliary signals for novel class learning but also effectively mitigates the problem of catastrophic forgetting, thereby enhancing the overall performance of incremental category learning.

## 4 EXPERIMENTS

### 4.1 EXPERIMENTAL DETAILS

#### 4.1.1 DATASET.

Following the standard protocol of Tao et al. (2020), we evaluate our method on CIFAR100, miniImageNet, and CUB200 under class-incremental settings. CIFAR100 and miniImageNet are each evaluated over 8 sessions with a 5-way 5-shot configuration, while CUB200 is evaluated over 10 sessions with a 10-way 5-shot setup.

#### 4.1.2 IMPLEMENTATION.

We adopt a two-phase training strategy for each dataset. For CUB200, the backbone consists of a pretrained ResNet-18 combined with a pretrained 12-layer ViT (192-dimensional embeddings, patch size 16), following (Tao et al., 2020; Tang et al., 2024). For CIFAR100, the backbone uses ResNet-12 and a 4-layer ViT (256-dimensional embeddings, patch size 16). For *mini*ImageNet, the same architecture as CIFAR100 is used, with the ViT patch size set to 12. In the first stage, the HDIN backbone is trained for 80 epochs with a batch size of 128. In the second stage, the AE-HMF module is trained with dataset-specific configurations. All models are optimized using SGD with momentum on two NVIDIA GeForce RTX 4090 GPUs, with a fixed random seed of 1 for reproducibility. The selected top-$k$ base classes are set to 2. More details are provided in Appendix A.1.

#### 4.1.3 EVALUATION PROTOCOL AND METRIC.

To comprehensively evaluate model performance, we adopt the following metrics. Accuracy in each session records the Top-1 accuracy computed over all classes seen up to the current session. Average Accuracy computes the mean accuracy across all sessions, reflecting overall performance and balance. Performance Decline Rate (Wang et al., 2024a) normalizes the decline and facilitates fair comparisons across models with different initial performance, i.e., DR $= (A_0 - A_N)/A_0$.

Table 1: 5-way 5-shot incremental learning results on CIFAR100. The best results are highlighted in bold, while the second-best results are underlined.

| Methods | Acc. in each session (%) | | | | | | | | | aACC↑ | DR↓ |
|---|---|---|---|---|---|---|---|---|---|---|---|
| | 0 | 1 | 2 | 3 | 4 | 5 | 6 | 7 | 8 | | |
| TOPIC (Tao et al., 2020) | 64.1 | 55.9 | 47.1 | 45.2 | 40.1 | 36.4 | 34.0 | 31.6 | 29.4 | 42.62 | 54.1 |
| CEC (Zhang et al., 2021) | 73.1 | 68.9 | 65.3 | 61.2 | 58.1 | 55.6 | 53.2 | 51.3 | 49.1 | 59.53 | 32.8 |
| CLOM (Zou et al., 2022) | 74.2 | 69.8 | 66.2 | 62.4 | 59.3 | 56.5 | 54.4 | 52.2 | 50.2 | 60.56 | 32.3 |
| FACT (Zhou et al., 2022a) | 74.6 | 72.1 | 67.6 | 63.5 | 61.4 | 58.4 | 56.3 | 54.2 | 52.1 | 62.24 | 30.2 |
| C-FSCIL (Yang et al., 2023) | 77.5 | 72.4 | 67.5 | 63.3 | 59.8 | 57.0 | 54.4 | 52.5 | 50.5 | 61.66 | 34.8 |
| SAVC (Song et al., 2023) | 78.8 | 73.3 | 69.3 | 64.9 | 61.7 | 59.2 | 57.1 | 55.2 | 53.1 | 63.63 | 32.6 |
| NC-FSCIL (Yang et al., 2023) | 82.5 | 76.8 | 73.3 | 69.7 | 66.2 | 62.9 | 61.0 | 59.0 | 56.1 | 67.50 | 32.0 |
| ALFSCIL (Li et al., 2024) | 80.8 | 77.9 | 72.9 | 68.8 | 65.3 | 62.2 | 60.0 | 57.7 | 55.2 | 65.81 | 31.7 |
| KRRM (Wang et al., 2024c) | 81.2 | 77.2 | 73.3 | 69.4 | 66.7 | 63.9 | 62.2 | 59.6 | 57.4 | 66.00 | 29.3 |
| CLOSER (Oh et al., 2024) | 75.7 | 71.8 | 68.3 | 64.6 | 61.9 | 59.3 | 57.5 | 55.4 | 53.3 | 63.09 | 29.6 |
| YourSelf (Tang et al., 2024) | 82.9 | 76.3 | 72.9 | 67.8 | 65.2 | 62.0 | 60.7 | 58.8 | 56.6 | 67.02 | 31.7 |
| CoDF (Wang et al., 2024b) | 86.0 | 81.0 | 76.6 | 72.4 | 69.0 | 66.2 | 64.2 | 62.2 | 60.0 | **70.8** | 30.2 |
| **Ours** | 84.0 | 79.5 | 75.8 | 71.9 | 68.5 | 66.0 | 63.9 | 62.0 | 59.7 | 69.87 | **28.9** |

## 4.2 MAIN RESULTS

We evaluate our method on CIFAR100, *mini*ImageNet, and CUB200, as reported in Tables 1, 2, and Appendix A.2, respectively. On *mini*ImageNet, our approach achieves an initial accuracy of 85.4% in Session 0 and maintains 61.0% by the final session, resulting in aACC of 71.06%. This performance clearly exceeds all ResNet-based baselines and is competitive with ViT-based approaches. Crucially, our method exhibits the lowest DR of 28.6%, indicating strong resistance to catastrophic forgetting. Although CoDF attains a slightly higher aACC, it depends on a costly MAE-based cognition pretraining stage. In contrast, our method avoids such self-supervised pretraining and provides explicit architectural disentanglement of common and specific features, making it more scalable to incremental settings. Moreover, unlike CoDF—which freezes the backbone and often yields overlapping decision boundaries for novel classes, our AE-HMF module actively adjusts novel-class boundaries to enhance novel-class separability.

Overall, under the setting of no large-scale pretraining or data augmentation techniques (Wang et al., 2024a), our method achieves the lowest DR across all benchmarks, highlighting an optimal balance between base-class representation and incremental stability. The combination of high accuracy and minimal degradation highlights the robustness and practicality of our method for FSCIL scenarios.

## 4.3 ABLATION STUDIES

To assess the effectiveness of the proposed HDIN and AE-HMF modules in the FSCIL framework, we conducted comprehensive ablation studies on the CIFAR100 dataset. Table 3 summarizes the results under various configurations, elucidating the individual contributions of each component.

First, ablation results demonstrate the role of the HDIN module's dual streams. When LSS and GGS are used alone, both performances are significantly lower than when the two are combined. The GGS stream in particular shows a much lower result, mainly because it relies on a four-layer ViT (see Appendix A.1) and requires more training data to achieve competitive performance. This suggests that the unified representation space is established through the bidirectional interaction between the ResNet-based fine-grained dis-

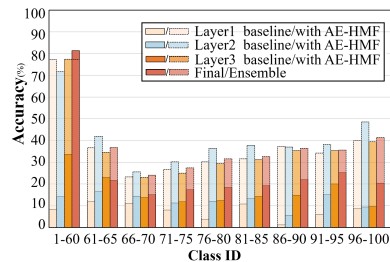

Figure 4: Ablation study of the AE-HMF Module. The solid bars and dashed bars represent the baseline and AE-HMF module, respectively.

Table 2: 5-way 5-shot incremental learning results on *mini*ImageNet. The best results are highlighted in bold, while the second-best results are underlined.

| Methods | Acc. in each session (%) | | | | | | | | | aACC↑ | DR↓ |
|---|---|---|---|---|---|---|---|---|---|---|---|
| | 0 | 1 | 2 | 3 | 4 | 5 | 6 | 7 | 8 | | |
| TOPIC (Tao et al., 2020) | 61.3 | 50.1 | 45.2 | 41.2 | 37.5 | 35.5 | 32.2 | 29.5 | 24.4 | 39.64 | 60.2 |
| CEC (Zhang et al., 2021) | 72.0 | 66.8 | 63.0 | 59.4 | 56.7 | 53.7 | 51.2 | 49.2 | 47.6 | 57.75 | 33.9 |
| CLOM (Zou et al., 2022) | 73.1 | 68.1 | 64.2 | 60.4 | 57.4 | 54.3 | 51.5 | 49.4 | 48.0 | 58.48 | 34.3 |
| FACT (Zhou et al., 2022a) | 72.6 | 69.6 | 66.4 | 62.8 | 60.6 | 57.3 | 54.3 | 52.2 | 50.5 | 60.70 | 30.4 |
| C-FSCIL (Yang et al., 2023) | 76.4 | 71.1 | 66.5 | 63.3 | 60.4 | 57.5 | 54.8 | 53.1 | 51.4 | 61.61 | 32.7 |
| SAVC (Song et al., 2023) | 81.1 | 76.1 | 72.4 | 68.9 | 66.5 | 63.0 | 59.9 | 58.4 | 57.1 | 67.05 | 29.6 |
| NC-FSCIL (Yang et al., 2023) | 84.0 | 76.8 | 72.0 | 67.8 | 66.4 | 64.0 | 61.5 | 59.5 | 58.3 | 67.82 | 30.6 |
| ALFSCIL (Li et al., 2024) | 81.3 | 76.0 | 71.0 | 66.5 | 63.5 | 60.0 | 56.9 | 54.8 | 53.3 | 64.90 | 34.2 |
| KRRM (Wang et al., 2024c) | 82.7 | 77.8 | 73.6 | 70.2 | 67.7 | 64.8 | 61.9 | 60.0 | 58.4 | 68.01 | 29.4 |
| CLOSER (Oh et al., 2024) | 76.0 | 71.6 | 68.0 | 64.7 | 61.7 | 58.9 | 56.2 | 54.5 | 53.3 | 62.77 | 29.9 |
| YourSelf (Tang et al., 2024) | 84.0 | 77.6 | 73.7 | 70.0 | 68.0 | 64.9 | 62.1 | 59.8 | 59.0 | 68.80 | 29.8 |
| CoDF (Wang et al., 2024b) | 88.0 | 81.4 | 78.0 | 73.8 | 70.3 | 67.1 | 64.3 | 62.3 | 60.9 | **71.79** | 32.8 |
| **Ours** | 85.4 | 80.2 | 76.2 | 73.0 | 70.4 | 67.0 | 64.1 | 62.2 | 61.0 | 71.06 | **28.6** |

Table 3: Ablation studies on CIFAR100. IGS represents the Information-driven Gaussian sampling module.

| HDIN | | AE-HMF | | | | Acc. in each session (%) | | | | | | | | | aAcc↑ |
|---|---|---|---|---|---|---|---|---|---|---|---|---|---|---|---|
| LSS | GGS | $\mathcal{L}_{cos}$ | $\mathcal{L}_{novel}$ | $\mathcal{L}_{global}$ | replay | 0 | 1 | 2 | 3 | 4 | 5 | 6 | 7 | 8 | |
| ✓ | | | | | | 75.9 | 71.4 | 67.3 | 63.5 | 60.4 | 57.4 | 55.2 | 53.2 | 51.1 | 61.7 |
| | ✓ | | | | | 48.2 | 45.6 | 42.9 | 40.4 | 38.3 | 36.4 | 34.6 | 33.2 | 31.9 | 39.1 |
| ✓ | ✓ | | | | | 84.0 | 79.3 | 75.0 | 70.6 | 67.3 | 64.2 | 61.9 | 59.7 | 57.5 | 68.9 |
| | ✓ | ✓ | | | IGS | 84.0 | 79.1 | 74.4 | 70.2 | 66.6 | 63.6 | 61.2 | 59.0 | 56.6 | 68.3 |
| | ✓ | ✓ | ✓ | | IGS | 84.0 | 75.8 | 72.8 | 67.2 | 64.5 | 62.2 | 61.0 | 59.4 | 56.6 | 66.6 |
| | ✓ | ✓ | | ✓ | IGS | 84.0 | 79.8 | 75.4 | 71.4 | 68.1 | 65.2 | 63.0 | 61.2 | 58.6 | 69.6 |
| | ✓ | ✓ | ✓ | ✓ | random | 84.0 | 79.6 | 75.6 | 71.5 | 68.2 | 65.6 | 63.3 | 61.2 | 58.6 | 69.7 |
| | ✓ | ✓ | ✓ | ✓ | IGS | **84.0** | **79.5** | **75.8** | **71.9** | **68.5** | **66.0** | **63.9** | **62.0** | **59.7** | **70.1** |

criminative pathway and the ViT-based global semantic abstraction pathway, thereby significantly enhancing feature discriminability and robustness.

Using HDIN alone—without AE-HMF or auxiliary loss terms as the baseline, we observe from Figure 4 that the introduction of AE-HMF facilitates effective multi-layer integration of base and novel class features (from fusion layer1 to fusion layer3). Notably, our method maintains stable performance on base classes and simultaneously achieves substantial improvements in novel class accuracy across incremental sessions. This balanced capability not only mitigates the effects of catastrophic forgetting but also enhances adaptability to newly introduced categories.

Further analysis of AE-HMF's loss components indicates that the optimal configuration combines $\mathcal{L}_{cos}$, $\mathcal{L}_{novel}$, and $\mathcal{L}_{global}$, achieving 59.7% accuracy in the 8th session and an average accuracy of 70.1%. Furthermore, we investigated the impact of the replay strategy in AE-HMF. Information-driven Gaussian sampling (IGS) slightly outperforms random sampling, indicating that selectively replaying samples based on mutual information facilitates more effective knowledge consolidation. In summary, the ablation results show the potential of the proposed modules in addressing the core challenges of class-incremental learning.

### 4.4 VISUALIZATION AND INTERPRETABILITY ANALYSIS

#### 4.4.1 CLASS ACTIVATION MAP (CAM) VISUALIZATIONS OF HDIN

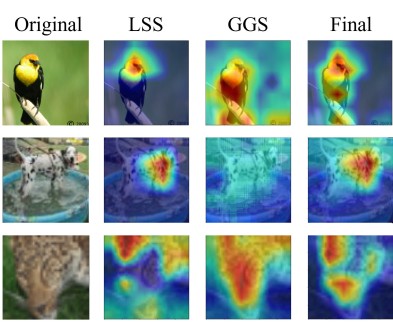

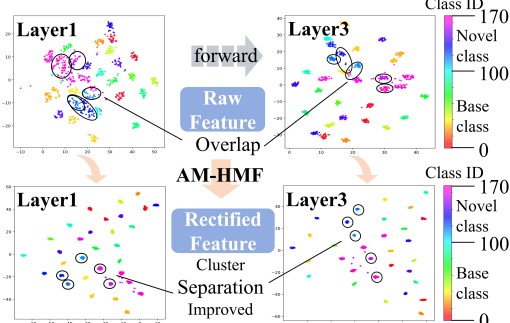

(a) Class Activation Map (CAM) visualizations of test set. From left to right: original input image, local-specific stream (LSS) activation, global-generic stream (GGS) activation, and the final integrated feature representation.

(b) T-SNE visualization of feature space optimization by AE-HMF. Top: Raw features exhibit interclass overlap across layers. Bottom: Rectifies feature spaces improves intra-class compactness and interclass separation.

Figure 5: Comparative visualization of model interpretability and feature space optimization

As shown in Figure 5a, we employ Grad-CAM (Selvaraju et al., 2017) to visualize the attention distribution of the Hierarchical Dual-stream Interaction Network (HDIN), highlighting the complementary roles of its two streams. The local-specific stream (LSS, second column) focuses on fine-grained details such as textures and structural features. In contrast, the global generic stream (GGS, third column) captures broader contextual cues like object shape and semantically relevant regions. The final fused representation (last column) integrates local and global features into a balanced attention map. These results validate the effectiveness of the dual-stream design in supporting interpretable feature disentanglement and simulate the human-like "attribute-commonality" cognitive process in few-shot class-incremental learning.

#### 4.4.2 T-SNE VISUALIZATION OF HIERARCHICAL MEMORY FUSION FOR FEATURE SPACE OPTIMIZATION.

As shown in Figure 5b, the t-SNE visualization (van der Maaten & Hinton, 2008) demonstrates that the AE-HMF mechanism significantly improved feature discriminability across all network layers. Compared to the baseline model, the optimized feature space substantially enhanced intra-class compactness and markedly increased inter-class separation. This hierarchical organization confirms that AE-HMF's memory fusion successfully preserves fine-grained details while simultaneously preventing feature drift across incremental tasks, resulting in a more structured and stable embedding space for few-shot class-incremental learning.

## 5 CONCLUSION

In this work, we propose a cognition-inspired framework for FSCIL that alleviates the core challenge of balancing base-class discriminability and novel-class adaptability. The framework integrates two complementary modules: the Hierarchical Dual-Stream Interaction Network, which captures multi-level visual representations via parallel ResNet and ViT pathways, and the Associative-Enhanced Hierarchical Memory Fusion, which aligns features across layers through memory-guided Gaussian sampling. Extensive experimental results demonstrate that our method consistently achieves significant performance improvement and low forgetting, with the lowest DR across all benchmarks under the setting of no large-scale pretraining or data augmentation techniques. In future work, we plan to extend this cognition-inspired framework to more practical few-shot continual learning scenarios characterized by sparse supervision, dynamic environments, and real-world constraints.

**Ethics Statement.** This work adheres to the ICLR Code of Ethics. Our study does not involve human subjects, personal data, or sensitive information. All datasets used are publicly available, properly cited, and employed in compliance with their intended usage policies. We have carefully considered potential ethical risks, including privacy, fairness, and possible misuse of the proposed methods. Our approach is designed for advancing research in Few-Shot Class-Incremental Learning, to improve generalization and mitigate catastrophic forgetting, and does not introduce foreseeable harmful applications. The authors confirm that there are no conflicts of interest or external sponsorship that could influence the outcomes of this research.

**Reproducibility Statement.** The implementation details, model configurations, and training procedures are provided in Section 4.1.2 and Appendix A.1. Full hyperparameters, ablation study setups, and dataset processing steps are outlined in Appendix A.1. While the code is not open-sourced at this time, all necessary details to reproduce the results, including theoretical assumptions, experimental setups, and data processing pipelines, are included in the supplementary materials.

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

APPENDIX

# A    MORE EXPERIMENTAL DETAILS

## A.1    DETAILED IMPLEMENTATION DETAILS

**CUB200:** In the first phase, initial learning rates are set to 0.005 for the ResNet and 0.01 for the ViT layers to train the backbone. In the second phase, the AE-HMF module is trained for 80 epochs on base classes and 40 epochs on novel classes with learning rate set to 0.1, with parameters: $\lambda_{\cos} = 0.2$, $\lambda_{\text{nov}} = 0.8$, and $\lambda_{\text{glo}} = 0.2$.

**CIFAR100:** In the first phase, the backbone is trained with an initial learning rate of 0.1. The AE-HMF module is trained for 30 epochs on base classes and 40 epochs on novel classes with a learning rate set to 0.1, with parameters: $\lambda_{\cos} = 0.5$, $\lambda_{\text{nov}} = 0.2$, and $\lambda_{\text{glo}} = 0.5$.

***mini*ImageNet:** The same settings as CIFAR100 in the first phase. The AE-HMF module is also trained under the same setting, but with $\lambda_{\text{glo}} = 0.6$.

## A.2    THE EXPERIMENT RESULT ON CUB200

The results of the CUB200 dataset are presented in Table **??**. While our method performs slightly weaker than CoDF (Wang et al., 2024b), with a marginal difference of only 0.5% in PD and 1.0% in DR, it is important to note several key factors that contextualize this gap. As discussed in the main text, CoDF is also a cognitively inspired method, but it requires approximately 2000 training epochs on the base session to achieve optimal performance. In contrast, our method is significantly more training-efficient.

Additionally, CUB-200 is a highly challenging fine-grained benchmark containing 200 bird species, where subtle inter-class variations make the task intrinsically difficult. A moderate performance gap in such a setting is therefore reasonable and expected. Nevertheless, our method still delivers competitive results—ranking second in both aACC and DR—while maintaining strong generalization without relying on large-scale pretraining or excessively large model capacity. To ensure a fair comparison, the ViT blocks used in the common stream of HDIN on CUB-200 follow the same configuration as ViT-T. In contrast, CoDF reports results with both ViT-T/16 and ViT-S/16, and its highest scores are obtained using the larger ViT-S backbone. To verify that our approach is similarly scalable, we further equip HDIN with ViT-S blocks. Under this setting, our method achieves state-of-the-art performance, demonstrating that the proposed architecture is not only competitive under equal backbone capacity but also capable of matching or surpassing the best reported results when scaled.

We further analyze the effect of each AE-HMF loss component. Table 5 presents the accuracy results during incremental learning for different loss combinations, while Figure 6 provides a more detailed view of the accuracy trends across classes in each incremental session, corresponding to the scenarios outlined in Table 5. When applying $\mathcal{L}_{\cos}$ alone, the model achieved baseline comparable performance in the first session but experienced a steady decline to 70.8% by the 10th task, indicating that feature alignment supports short-term stability but lacks sufficient constraints for long-term knowledge integration. In contrast, adding only Lnovel significantly improved novel class recognition but accelerated forgetting of base classes, leading to overall performance degradation due to optimization imbalance—reflecting the stability-plasticity dilemma. The inclusion of Lglobal improved global knowledge transfer and yielded higher early-session accuracy, yet failed to mitigate forgetting, suggesting its limited effect on novel class adaptation. Ultimately, combin ing all three losses within AE-HMF yielded a comprehensive model configuration that consistently outperformed all other settings, achieving an average accuracy of 76.66% and reaching 73.0% in the 10th task. These results underscore that the joint loss strategy not only enhances novel cate gory learning but also effectively alleviates catastrophic for getting, thereby improving the model's overall stability and adaptability in incremental learning.

Table 4: 10-way 5-shot incremental learning results on CUB200. The best results are highlighted in bold, while the second-best results are underlined.

| Methods | Acc. in each session (%) | | | | | | | | | | | aACC↑ | DR↓ |
|---------|------|------|------|------|------|------|------|------|------|------|------|-------|------|
| | 0 | 1 | 2 | 3 | 4 | 5 | 6 | 7 | 8 | 9 | 10 | | |
| TOPIC (Tao et al., 2020) | 68.7 | 62.5 | 54.8 | 50.0 | 45.2 | 41.4 | 38.4 | 35.4 | 32.2 | 28.3 | 26.3 | 43.93 | 61.7 |
| CEC (Zhang et al., 2021) | 75.9 | 71.9 | 68.5 | 63.5 | 62.4 | 58.3 | 57.7 | 55.8 | 54.8 | 53.5 | 52.3 | 61.33 | 31.1 |
| CLOM (Zou et al., 2022) | 79.6 | 76.0 | 72.9 | 69.8 | 67.1 | 65.6 | 63.9 | 62.6 | 60.6 | 60.3 | 59.6 | 67.09 | 25.1 |
| FACT (Zhou et al., 2022a) | 75.9 | 73.2 | 70.8 | 66.1 | 65.6 | 62.2 | 61.7 | 59.8 | 58.4 | 57.9 | 56.9 | 64.41 | 25.0 |
| SAVC (Song et al., 2023) | 81.9 | 77.9 | 75.0 | 70.2 | 70.0 | 67.0 | 66.2 | 65.3 | 63.8 | 63.2 | 62.5 | 69.36 | 23.7 |
| ALFSCIL (Li et al., 2024) | 79.8 | 76.5 | 73.1 | 69.0 | 67.6 | 64.8 | 63.5 | 62.3 | 60.8 | 60.2 | 59.4 | 67.00 | 25.6 |
| KRRM (Wang et al., 2024c) | 79.5 | 76.1 | 73.1 | 69.3 | 68.0 | 65.9 | 64.5 | 63.8 | 62.2 | 62.0 | 61.0 | 67.76 | 23.3 |
| CLOSER (Oh et al., 2024) | 79.4 | 75.9 | 73.5 | 70.5 | 69.2 | 67.2 | 66.7 | 65.7 | 64.0 | 64.0 | 63.6 | 69.06 | 19.9 |
| Yourself (Tang et al., 2024) | 83.4 | 77.0 | 75.3 | 72.2 | 69.9 | 66.8 | 66.0 | 65.6 | 64.1 | 64.5 | 63.6 | 69.80 | 24.9 |
| CODF[1] (Wang et al., 2024b) | 82.5 | 79.6 | 76.8 | 72.2 | 72.3 | 69.5 | 69.0 | 68.2 | 67.1 | 67.1 | 67.0 | 71.88 | 19.4 |
| CoDF[2] (Wang et al., 2024b) | 87.6 | 85.3 | 83.5 | 80.5 | 80.5 | 78.0 | 77.7 | 77.7 | 76.8 | 76.8 | 76.3 | 80.05 | 12.6 |
| **Ours**[1] | 84.5 | 81.8 | 79.9 | 78.0 | 76.5 | 74.6 | 74.2 | 74.3 | 73.0 | 73.5 | 73.0 | 76.66 | 13.6 |
| **Ours**[2] | 87.3 | 85.7 | 85.1 | 82.9 | 81.7 | 80.3 | 80.1 | 80.1 | 79.5 | 79.6 | 79.4 | **81.97** | **9.0** |

[1] Backbone incorporates ViT-T/16 blocks pre-trained on the ImageNet dataset.

[2] Backbone incorporates ViT-S/16 blocks pre-trained on the ImageNet dataset.

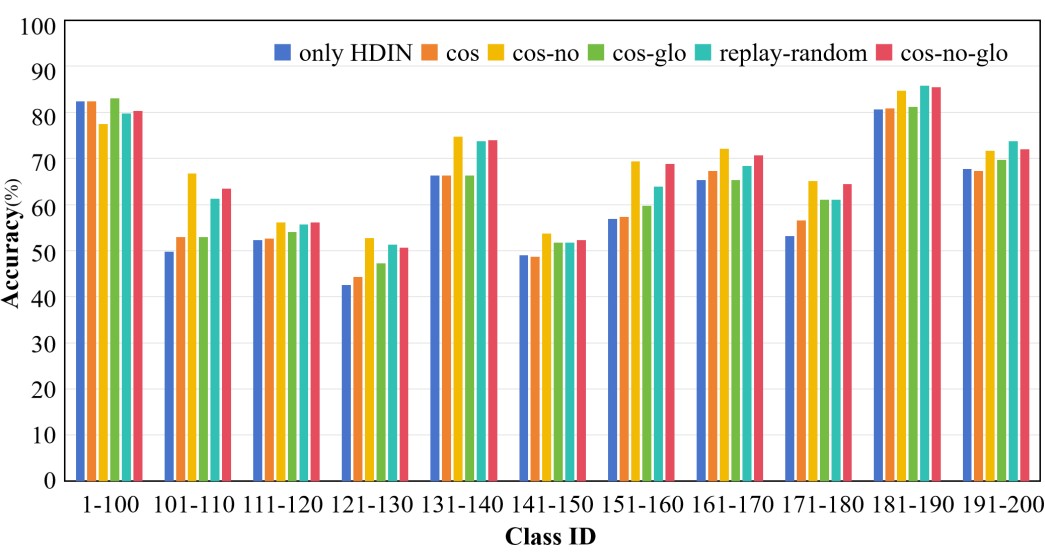

Figure 6: Ablation study of the AE-HMF Module.

### A.3 ANALYSIS OF THE IMPACT OF DATA EXPANSION AND PRETRAINED MODELS

A number of recent approaches have demonstrated that leveraging pretrained large models and expanding training datasets can significantly enhance the performance of FSCIL methods. As shown in Table 6, Wang et al. (2024a) highlights that augmenting the training data for base classes is a straightforward and powerful strategy to address the challenges posed by FSCIL, particularly in scenarios where the initial dataset is relatively small. Data expansion techniques, such as the use of mixup (Pinto et al., 2022; Zhang et al., 2017) and other augmentation methods (Balestriero et al., 2022; Hao et al., 2023), have also been commonly employed to improve performance in similar settings. Additionally, pretrained foundation models like CLIP (Radford et al., 2021) have been shown to enhance performance by providing a richer feature representation space, as evidenced by the results of methods such as CoCoOp (Zhou et al., 2022c) and Prompt (Yoon et al., 2023).

Table 5: Ablation studies of the AE-HMF module on CUB200. IGS represents the Information-driven Gaussian sampling module.

| $\mathcal{L}_{cos}$ | $\mathcal{L}_{novel}$ | $\mathcal{L}_{global}$ | replay | 0 | 1 | 2 | 3 | 4 | 5 | 6 | 7 | 8 | 9 | 10 | aAcc↑ |
|---|---|---|---|---|---|---|---|---|---|---|---|---|---|---|---|
| | | | | 84.2 | 81.4 | 79.0 | 76.3 | 74.8 | 72.8 | 72.0 | 71.9 | 70.3 | 70.7 | 70.4 | 74.89 |
| ✓ | | | IGS | 84.5 | 81.6 | 79.4 | 76.7 | 75.1 | 73.1 | 72.5 | 72.2 | 70.8 | 71.2 | 70.8 | 75.26 |
| ✓ | ✓ | | IGS | 84.5 | 80.4 | 79.3 | 77.1 | 75.1 | 73.0 | 72.8 | 73.0 | 71.9 | 72.3 | 72.0 | 75.58 |
| ✓ | | ✓ | IGS | 84.5 | 81.8 | 79.6 | 77.0 | 75.5 | 73.8 | 73.4 | 73.1 | 72.0 | 72.2 | 71.9 | 75.89 |
| ✓ | ✓ | ✓ | random | 84.0 | 81.2 | 79.1 | 76.7 | 75.4 | 73.8 | 73.6 | 73.3 | 71.9 | 72.6 | 72.1 | 75.79 |
| ✓ | ✓ | ✓ | IGS | **84.5** | **81.8** | **79.9** | **78.0** | **76.5** | **74.6** | **74.2** | **74.3** | **73.0** | **73.5** | **73.0** | **76.66** |

The two leftmost header groups: **AE-HMF** spanning $\mathcal{L}_{cos}$, $\mathcal{L}_{novel}$, $\mathcal{L}_{global}$, replay; and **Acc. in each session (%)** spanning sessions 0–10.

Table 6: Comparison of 5-way 5-shot Incremental Learning Results with Pretrained Models and Data Expansion on *mini*ImageNet. The symbols † and ⋄ signify models derived from the CLIP (Radford et al., 2021) and used data expansion, respectively.

| Methods | Backbone(Para.) | 0 | 1 | 2 | 3 | 4 | 5 | 6 | 7 | 8 | aACC↑ | DR↓ |
|---|---|---|---|---|---|---|---|---|---|---|---|---|
| CoCoOp[†] (Zhou et al., 2022c) | ViT-B/16($\sim$88M) | 94.2 | 93.4 | 90.2 | 86.6 | 85.2 | 84.1 | 83.8 | 83.6 | 82.7 | 87.09 | 12.2 |
| Prompt[†] (Yoon et al., 2023) | ViT-B/16($\sim$88M) | 95.4 | 94.4 | 93.4 | 93.1 | 92.1 | 91.4 | 91.4 | 90.7 | 90.0 | 92.43 | 5.7 |
| Appromaxtion[†] (Wang et al., 2024a) | ViT-B/16($\sim$88M) | 93.2 | 93.2 | 91.2 | 90.5 | 90.5 | 90.0 | 89.1 | 89.1 | 89.0 | 90.64 | 4.4 |
| Appromaxtion⋄ (Wang et al., 2024a) | ResNet18(11.2M) | 84.2 | 81.3 | 77.0 | 74.6 | 71.7 | 68.7 | 66.0 | 63.8 | 63.2 | 72.29 | 26.2 |
| Appromaxtion (Wang et al., 2024a) | ResNet18(11.2M) | 73.2 | 68.3 | 64.0 | 60.6 | 57.7 | 54.7 | 52.0 | 49.8 | 48.2 | 58.73 | 34.2 |
| **Ours** | HDIN(15.8M) | 85.4 | 80.2 | 76.2 | 73.0 | 70.4 | 67.0 | 64.1 | 62.2 | 61.0 | 71.06 | 28.6 |

In contrast, our proposed approach does not rely on either data expansion or pretrained models. This design decision serves several key purposes. First, by not using pretrained models, our method avoids the potential pitfalls of domain shift and the reliance on large-scale models that may not generalize well to specific FSCIL tasks. Additionally, the absence of data expansion techniques ensures that the model's performance is not artificially inflated by external data sources, allowing for a more realistic and transparent evaluation of its generalization capabilities within the constraints of the original dataset. This decision also provides a more efficient model, as we do not incur the computational overhead associated with training or fine-tuning large pretrained models or generating augmented data.

Despite the inherent advantages of our approach, we acknowledge that data expansion and pretrained models have the potential to further improve performance. In future work, we may explore incorporating these strategies to investigate whether they lead to more robust and scalable solutions for FSCIL. Specifically, using pretrained models could allow our approach to benefit from learned representations that generalize across a wider range of tasks, while data augmentation could help alleviate issues related to limited data in incremental learning settings. However, these enhancements would require careful consideration of the trade-offs, particularly in terms of computational complexity and model interpretability.

## A.4 ANALYSIS OF THE HYPER-PARAMETERS

In order to study the effect of different parameter values, we conduct experiments on key hyper-parameters $\lambda_{cos}$, $\lambda_{nov}$, $\lambda_{glo}$ and $k$. Experiments are conducted on CIFAR-100 and results are shown in Figure 7.

- **The effect of $\lambda_{cos}$.** This parameter controls the weighting of the cosine-alignment loss $\mathcal{L}cos$. As illustrated in Figure 7(a), performance remains remarkably consistent as $\lambda cos$ varies from 0.01

to 0.9. This observation suggests that inter-layer feature alignment functions predominantly as a benign regularizing influence, which does not perturb the underlying stability–plasticity balance.

- **The effect of $\lambda_{\text{nov}}$.** The weighting coefficient for novel-class cross-entropy $\mathcal{L}$nov, controlling the intensity of fitting to novel classes. As illustrated in Figure 7(b), $\lambda_{\text{nov}}$ exhibits a strong sensitivity. Moderate values ($\leq 0.3$) provide relatively stable performances while excessively large $\lambda_{\text{nov}}$ ($\geq 0.7$) leads to pronounced performance degradation on both metrics, indicating that overly strong novel-class supervision disrupts previously learned decision boundaries.
- **The effect of $\lambda_{\text{glo}}$.** This parameter controls the weighting of the global classification loss $\mathcal{L}$glo, employed to stabilize overall class boundaries and prevent excessive shifts in learned boundaries caused by novel-class fitting. As illustrated in Figure 7(c), $\lambda_{\text{glo}}$ stable behavior across all tested values. This suggests that enforcing global decision consistency provides beneficial regularization.
- **The effect of $k$.** This parameter specifies the number of semantic neighbors used in AE-HMF. As illustrated in Figure 7(d), a value of $k = 1$ results in the weakest stability due to insufficient semantic evidence, while larger $k$ values (5 or 10) introduce redundant neighbors that provide limited additional benefit. Consequently, we set $k = 2$ as the default.

| (a) The effect of $\lambda_{\text{cos}}$ | (b) The effect of $\lambda_{\text{nov}}$ | (c) The effect of $\lambda_{\text{glo}}$ | (d) The effect of $k$ |

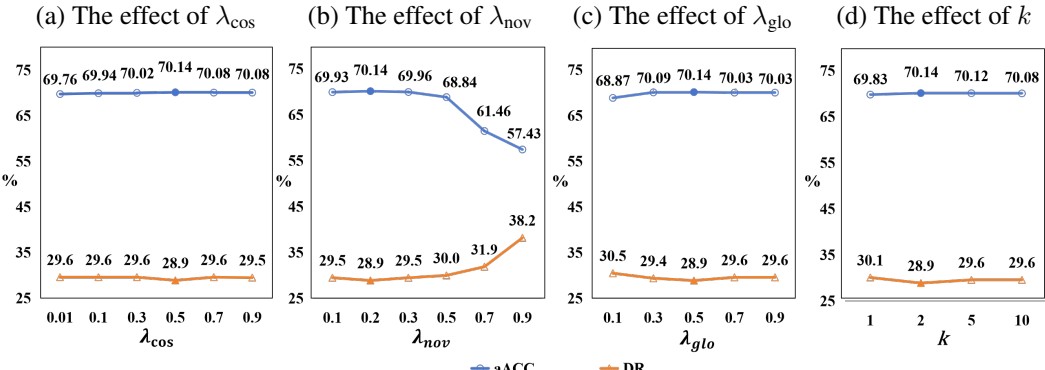

Figure 7: Ablation studies on the impact of key hyper-parameters $\lambda_{\text{cos}}$, $\lambda_{\text{nov}}$, $\lambda_{\text{glo}}$ and $k$, which are conducted on CIFAR-100. We report both aAcc and DR results in the figure. Best viewed in color. The final selected configuration is denoted by solid markers.

## B  THE PSEUDOCODE OF THE PROCEDURE OF INFORMATION-DRIVEN GAUSSIAN SAMPLING

The pseudocode Algorithm 1 summarizes the complete procedure for constructing multivariate Gaussian priors for novel classes during each incremental session.

## C  USE OF LLMS

**Scope of use.** (1) Sentence-level copy-editing for clarity, concision, and tone; (2) restructuring paragraphs for readability; (3) fixing minor style inconsistencies (e.g., tense, article use, punctuation). When interacting with the LLM, we explicitly instructed: "Do not introduce new claims, data, citations, or equations; preserve technical meaning exactly; keep numbers, symbols, and references unchanged."

**Non-use.** The LLM was not used to design experiments, implement algorithms, tune hyperparameters, compute results, write proofs, or interpret findings.

**Quality control and accountability.** All LLM-edited passages were reviewed line-by-line by the authors, and any suggestions that altered technical meaning were rejected. All quantitative results, equations, and claims were verified against our logs and source code. The authors take full responsibility for the final content of the paper.

---

**Algorithm 1** Information-Driven Gaussian Sampling for Novel Classes

---

**Require:** Novel support set $\mathcal{D}_{\text{novel}} = \{(x_i, y_i)\}_{i=1}^{N_{\text{novel}}}$ with $y_i \in \mathcal{C}_{\text{novel}}$, neighborhood size $k$ and regularization $\varepsilon$ hyperparameters, frozen HDIN backbone $h^{(l)}(\cdot)$ for layer-wise feature extraction and layers number $L$

**Ensure:** Updated `MVN_distributions` $= \{(\mu_c^{(l)}, \widehat{\Sigma}_c^{(l)})\}_{c \in \mathcal{C}_{\text{novel}}}^{l \in L}$, containing per-layer multivariate Gaussian models for all classes

---

1: **Initialize with frozen base-class statistics** ▷ Pre-computed during base training phase
2: **for each** base class $c \in \mathcal{C}_{\text{base}}$ **and** layer $l \in L$ **do**
3:      $\mathcal{X}_c \leftarrow \{h^{(l)}(x) \mid x \in \mathcal{D}_{\text{base}}, y = c\}$
4:      $\mu_c^{(l)} \leftarrow \text{mean}(\mathcal{X}_c)$
5:      $\Sigma_c^{(l)} \leftarrow \text{cov}(\mathcal{X}_c)$
6: **end for**

7: **Process novel classes incrementally**
8: **for each** novel class $c' \in \mathcal{C}_{\text{novel}}$ **do**
9:      **for each** layer $l \in L$ **do**
10:          **Compute empirical statistics from few-shot samples**
11:          $\mathcal{X}_{c'} \leftarrow \{h^{(l)}(x) \mid x \in \mathcal{D}_{\text{novel}}, y = c'\}$
12:          $\mu_{c'}^{(l)} \leftarrow \text{mean}(\mathcal{X}_{c'})$
13:          $\Sigma_{c'}^{(l)} \leftarrow \text{cov}(\mathcal{X}_{c'})$

14:          **Compute MI-based similarity matrix**
15:          **for each** base class $c \in \mathcal{C}_{\text{base}}$ **do**
16:             $\mathcal{N}_{c'} \leftarrow \mathcal{N}(\mu_{c'}^{(l)}, \Sigma_{c'}^{(l)}), \mathcal{N}_c \leftarrow \mathcal{N}(\mu_c^{(l)}, \Sigma_c^{(l)})$
17:             $S[c', c] \leftarrow I(\mathcal{N}_{c'}; \mathcal{N}_c) = H(\mathcal{N}_{c'}) + H(\mathcal{N}_c) - H(\mathcal{N}_{c'}, \mathcal{N}_c)$      ▷ Eq. (14)
18:          **end for**

19:          **Construct regularized Gaussian prior**
20:          $\mathcal{C}_k \leftarrow \text{topk}(S[c', :], k)$
21:          $\widehat{\Sigma}_{c'}^{(l)} \leftarrow \frac{1}{k} \sum_{b \in \mathcal{C}_k} \Sigma_b^{(l)} + \varepsilon I$      ▷ Eq. (16)
22:          `MVN_distributions[c'][l]` $\leftarrow \mathcal{N}(\mu_{c'}^{(l)}, \widehat{\Sigma}_{c'}^{(l)})$
23:      **end for**
24: **end for**

25: **return** `MVN_distributions`

---

