# OpenReview forum: "Few-Shot Class-Incremental Learning based on Hierarchical Dual-Stream Interaction and Associative Memory Fusion"
_ICLR.cc/2026/Conference — Submitted to ICLR 2026_

### Official Review · Reviewer_uqHE · 2025-10-27

**Soundness:** 3
**Presentation:** 2
**Contribution:** 2
**Rating:** 4
**Confidence:** 4

**Summary:**

The paper proposes a new Few-Shot Class-Incremental Learning method by adopting a Hierarchical Dual-Stream Interaction Network (HDIN), which consists of a ResNet-based local stream and a ViT-based global stream. Associative-Enhanced Hierarchical Memory Fusion (AE-HMF) module performs cross-layer fusion and information-driven Gaussian sampling to enhance long-term memory retention and semantic stability. Experiments are conducted on CIFAR-100, miniImageNet, and CUB200 datasets.

**Strengths:**

HDIN has a well-motivated architecture, which decouples local and global cues and recombines them via a reasonable attention fusion strategy. AE-HMF’s hierarchical cross-layer fusion with mutual information-guided Gaussian sampling is described with clear mathematical details. HDIN reports a consistently low decline rate on CIFAR-100 and miniImageNet.

**Weaknesses:**

Weakness

1. Figures 1–3 do not clearly convey how modules connect or how information flows. The relationships and contribution paths are hard to follow. The statement around Lines 203–205 is also unclear and should be rewritten for precision.

2. The information-driven Gaussian sampling needs more details to clarify.

3. Top-k selection of base classes via MI can be fragile to few-shot covariance. The paper lacks sensitivity studies. It is unclear how the training samples distilled by the sampling strategy are incorporated into the training stage.

4. The quantitative performance of the proposed HDIN is not significant. On CUB200, CoDF outperforms HDIN in both aACC and DR.

5. Given the dual-stream and fusion complexity and the claim of better efficiency than CoDF, the proposed HDIN needs more quantitative evidence (e.g. FLOPs, training and inference time, GPU memory, parameter counts).

**Questions:**

For general suggestions, please refer to the weakness part. Here are several specific points to improve the article:

1. Revise Figures 1-3 to specify the workflow of HDIN, especially the interaction between the proposed modules. Add necessary details, such as what the dotted line indicates.

2. Are the empirical mean and covariance matrix fixed after the first round of computation? Regarding the description in lines 332-335, better to specify the update rule and provide a short pseudocode block if possible.

3. Study how sensitive HDIN results are to the selection of k in IGS, for example, report performance vs. k for MI-guided sample selection. If possible, discuss robustness under higher variance.

4. Please provide training and inference time, GPU memory, and parameter counts for each module vs. methods for comparison, particularly CoDF.

5. To align with the description of the cognition-driven learning paradigm in Figure 1, include Figure 6 (currently in the supplement) as supporting evidence referenced in the main manuscript.

---

> ### Author Response · Authors · 2025-11-24
> **Response to Reviewer uqHE (Part 1/2)**
>
> > **W1 & Q1: Revise Figures 1–3 to specify the workflow.**
>
> **A1:** We appreciate the reviewer’s feedback. We agree that the readability of Figures 1-3 could be further enhanced. In the revised version, we have revised unclear annotations and redrawn these figures to explicitly exhibit the workflow between modules and provided more details to convey the core logic of our methodology. We have also rewritten the statement around Lines 203-205 in the revised manuscript.
>
> > **W2: The information-driven Gaussian sampling needs more details to clarify.**
>
> **A2:** Thank you for your valuable suggestion. In the revised manuscript, we have added a more detailed description. Specifically, we have clarified the process of selecting informative base classes based on the mutual information matrix, aggregating and regularizing their covariance statistics, and forming per-layer Gaussian generators that support distribution-consistent sampling for novel categories. The newly added explanation is highlighted in blue in the revised PDF.
>
>
>
> > **Q2: Are the empirical mean and covariance fixed after the first computation? Please specify the update rule and give pseudocode.**
>
> **A3:** Yes, the empirical mean and covariance matrix are fixed after the first round of computation. This is essential to ensure stability and prevent feature drift when novel classes lack sufficient samples. Based on the mutual information matrix, we select the top-k base classes with the highest scores and leverage their mean covariance information to update the enhanced distribution of the novel class, thereby constructing a multi-granularity Gaussian distribution across each layer to generate enhanced samples for training the AE-HMF module. It enables continuous adaptation while mitigating catastrophic forgetting. We further introduce more details about IGS, see Section 3.4. And we provided a relative pseudocode block about this module in Appendix B. The pseudocode is available at: https://imgur.com/a/cxZouaD.
>
> > **W3 & Q3: Sensitivity of HDIN to choice of k in IGS and robustness under higher variance.**
>
> **A4:** We thank the reviewer for raising the important question regarding the sensitivity of the hyperparameter $k$. We have conducted a sensitivity analysis, which is now included as Fig. 7(d) in the Appendix of the revised PDF.
>
> As summarized in the Table below, performance remains relatively stable across a moderate range of $k$ values
>
> k=1 generally yields the weakest performance due to insufficient semantic context.
>
> k=2 provides a robust trade-off and is adopted as our default setting.
>
> k∈{5,10} leads to marginal gains in some scenarios, but often introduces redundant or noisy neighbors.
>
> Table 1: Ablation study on the hyperparameter $k$ in IGS.
>
> | $k$  |   1   |   2   |   5   |  10   |
> | :--: | :---: | :---: | :---: | :---: |
> | aACC | 69.83 | **70.14** | 70.12 | 70.08 |
> |  DR  | 30.1  | **28.9** | 29.6  | 29.6  |
>
> We construct covariance priors independently for each network layer rather than relying on a single global estimate. Additionally, we add to all covariance matrices and average covariances across the top-k selected neighbors, which ensures numerical stability. Enhanced samples generated using the regularized covariances are mixed with real support samples during training. These samples are optimized under a multi-task objective (combining cosine alignment and cross-entropy loss), ensuring that synthetic features refine—rather than replace—the real prototype distribution.
>
> Even when sampling under high variance, the feature rectification stage in AE-HMF—coupled with the joint training objective—effectively pulls synthetic features toward the manifold of real supports. This correction mechanism empirically prevents catastrophic class drift.  Taken together, these strategies ensure that our approach remains robust to the choice of k and performs reliably even under high-variance conditions.

---

> ### Author Response · Authors · 2025-11-24
> **Response to Reviewer uqHE (Part 2/2)**
>
> > **W4: Poor Performance increase on CUB200.**
>
> **A5:** To ensure that model sizes are on a comparable basis, the ViT blocks used in the common stream of HDIN follow the same configuration as ViT-T. CoDF reports results with ViT-T/16 and ViT-S/16; some of their best performances come from a larger backbone. We agree that this may have led to an underestimation of our full potential in the initial submission. To demonstrate that our method also has the potential to reach the best-reported performance, we adopt ViT-S blocks in the HDIN backbone. Our method’s performance improves (Table 1), which indicates that **the proposed architectural idea is compatible with larger backbones and can reach SOTA under that setting**.
>
> Table 2: Performance comparison with CoDF on CUB200 across different backbone architectures.
>
> |  Method  |          Backbone           |    0     |    1     |    2     |    3     |    4     |    5     |    6     |    7     |    8     |    9     |    10    |   aACC↑   |   DR↓   |
> | :------: | :-------------------------: | :------: | :------: | :------: | :------: | :------: | :------: | :------: | :------: | :------: | :------: | :------: | :-------: | :-----: |
> |   CoDF   |       ViT-T/16(5.6M)        |   82.5   |   79.6   |   76.8   |   72.2   |   72.3   |   69.5   |   69.0   |   68.2   |   67.1   |   67.1   |   67.0   |   71.88   |  19.4   |
> |   CoDF   |       ViT-S/16(21.6M)       |   87.6   |   85.3   |   83.5   |   80.5   |   80.5   |   78.0   |   77.7   |   77.7   |   76.8   |   76.8   |   76.3   |   80.05   |  12.6   |
> |   Ours   |   HDIN w/ ViT-T/16(17.3M)   |   84.5   |   81.8   |   79.9   |   78.0   |   76.5   |   74.6   |   74.2   |   74.3   |   73.0   |   73.5   |   73.0   |   76.66   |  13.6   |
> | **Ours** | **HDIN w/ ViT-S/16(33.8M)** | **87.3** | **85.7** | **85.1** | **82.9** | **81.7** | **80.3** | **80.1** | **80.1** | **79.5** | **79.6** | **79.4** | **81.97** | **9.0** |
>
>
>
> > **W5 & Q4: Provide training & inference time, GPU memory, and parameter counts vs methods (esp. CoDF)**
>
> **A6:** We appreciate the reviewer’s concern. Since our work does not primarily focus on designing lightweight network architectures or structural paradigms, we did not initially provide a detailed analysis and comparison of model complexity, memory efficiency, and computational efficiency. In view of your concerns about these aspects of our method, we have added corresponding experimental results.
>
> Table 3:  Comparison of model complexity and computational efficiency with CoDF on CUB200.
>
> | Method | Training phase | Total/Training Param | FLOPs  | Training Time | Inference Time（Average Sample） |
> | :----: | :------------: | :------------------: | :----: | :-----------: | :------------------------------: |
> |  CoDF  | Cognitive task |     23.7M/23.7M      | 1.59G  |   233.91min   |             0.0171 s             |
> |  Our   |  First-phase   |     17.3M/17.3M      | 3.08 G |   26.64min    |             0.0953s              |
>
> From the table, we can observe that the inference FLOPs and latency are indeed higher, which is mainly caused by the additional computations introduced to realize inter-layer interaction and associative modeling. However, our primary concern lies in the high cost of CoDF’s cognitive pretraining: its training time for the cognitive stage is nearly nine times longer than that of our single-stage training.
>
> Table 4: Parameter distribution across modules on CUB200.
>
> |            Module            | Parameter |
> | :--------------------------: | :-------: |
> |            Total             |   22.4M   |
> |             HDIN             |   17.1M   |
> |              fc              |   0.1M    |
> | feature_rectification.layer1 |   1.3M    |
> | feature_rectification.layer2 |   1.5M    |
> | feature_rectification.layer3 |   2.1M    |
> |       fc_layer.layer1        |   51.2K   |
> |       fc_layer.layer2        |   64.0K   |
> |       fc_layer.layer3        |   89.6K   |
> |       layer_attention        |   2.1K    |
>
> During incremental sessions, the HDIN backbone itself is frozen, we only fine-tune the `feature_rectification` and `layer_attention` modules, which minimizes computational overhead, GPU memory usage, and tuning burden. Additional memory costs primarily arise from the MVN distribution dictionary in AE-HMF (storing per-class and per-layer statistics) which scale linearly with the number of classes.  No rehearsal buffers, generative replay models are used. This significantly reduces tuning cost and prevents catastrophic forgetting.
>
>
>
> > **Q5: Include Figure 6 (currently in supplement) in the main manuscript.**
>
> **A7:** We agree that placing Figure 6 in the main text better echoes our original motivation, improves the overall readability of the paper, and helps to more clearly highlight the method’s motivation and contributions. Accordingly, we have moved Figure 6 back into the main manuscript.

---

> ### Comment · Reviewer_uqHE · 2025-11-26
>
> The authors have addressed the concerns regarding clarity, robustness, and performance compatibility. The revised workflow descriptions and the inclusion of pseudocode resolve the ambiguity in the initial submission. The additional experiments using the ViT show that the proposed method can outperform the baseline when architectures are aligned. However, the provided efficiency metrics confirm the initial concern regarding computational complexity, while the method achieves significantly faster training times, it does so at the cost of inference efficiency, exhibiting notably higher latency and FLOPs compared to CoDF. Other than this limitation, the revisions are satisfactory.

---

> > ### Author Response · Authors · 2025-11-26
> > **Thanks to Reviewer uqHE**
> >
> > We greatly appreciate your valuable suggestions regarding clarity, robustness, and performance compatibility, which have been instrumental in improving our manuscript. We are also grateful for your recognition of the improvements made in the revised version and for updating the score. Regarding the limitation in inference efficiency, we will continue to explore and optimize this aspect in our future work. Once again, we sincerely appreciate your time and professional guidance!

---

### Official Review · Reviewer_xph2 · 2025-11-01

**Soundness:** 2
**Presentation:** 2
**Contribution:** 2
**Rating:** 2
**Confidence:** 5

**Summary:**

The paper addresses the FSCIL problem and proposes a cognition-inspired framework which uses a hierarchical Dual-Stream Interaction Network (HDIN), consisting of a CNN and a ViT, and associative-Enhanced Hierarchical Memory Fusion (AE-HMF) which uses Gaussian samples and prototypes for reducing forgetting. Experiments on standard benchmarks show SOTA performance.

**Strengths:**

1) The paper is clearly written and the cognition inspired motivation is sound.
2) The proposed method shows SOTA performance on multiple benchmark FSCIL datasets.
3) Pairing local (CNN) and global (ViT) cues can help genearlize from limited data available in FSCIL.

**Weaknesses:**

1) The novelty of the paper is limited. There have been a large number of FSCIL methods that are cognition inspired, or use prototypes and Gaussian pseudo-samples. How is this paper contributing further than the large literature of FSCIL methods? Some examples of older FSCIL works that used these ideas: [a,b]
2) The method combines a large number of modules with many hyperparameters, making it more complex compared to prior methods. Have the authors analyzed the model compexity, the memory efficiency and computational efficiency? The gain in accuracy compared to many prior methods seems to be minimal (and less than CoDF). Does such a gain in accuracy warrant such a complex method that might incur more computational burden and require tuning more hyperparameters?
3) It was stated in the appendix that for CUB-200, the paper used pre-trained ImageNet features on both the CNN and the ViT. This should be clarified in the main paper. The paper should also compare against simpler works that use pre-trained ImageNet features and achieve SOTA performance. e.g. [b].
4) Gaussian sampling around prototypes risks class drift or over-smoothing of fine-grained classes. What variance schedule, class-adaptive covariance, and safeguards against prototype collapse are used?

[a] FearNet: Brain-Inspired Model for Incremental Learning, ICLR 2018.
[b] Ayub, A., & Fendley, C. (2022). Few-shot continual active learning by a robot. Advances in Neural Information Processing Systems, 35, 30612-30624.

**Questions:**

Please see the weaknesses section for my questions.

---

> ### Author Response · Authors · 2025-11-24
> **Response to Reviewer xph2 (Part 1/4)**
>
> > **Q1: The novelty is limited. Many cognition-inspired or prototype/Gaussian-based FSCIL works exist (e.g., [a,b]). How is this work different?**
>
> **A1:** Thank you very much for pointing out earlier cognition-inspired and prototype/Gaussian-based FSCIL works such as FearNet [a] and FoCAL [b]. We completely agree that these directions share conceptual similarities with ours, and we appreciate the chance to clarify what specific challenge our work aims to address and how our formulation differs.
>
> Rather than introducing a single new ingredient, our contribution lies in **re-examining FSCIL from a different angle**—focusing on two failure modes that existing cognition-inspired or Gaussian-sampling approaches do not directly target:
> (1) **intra-class variance collapse**, and (2) **decision-boundary drift** under extremely limited novel-class examples.
> This shift of perspective is what guided the architectural and sampling designs in our method.
>
> To address these overlooked challenges, we **reformulate FSCIL as a two-stage cognitive process**:
>
> - Rapid disentangled perception through a dual-stream, multi-granularity backbone (HDIN), which keeps common and specific features separated across layers and enables bidirectional local–global interaction. Existing single-stream backbones fuse features only at the output level and therefore cannot prevent fine-grained variance collapse as new classes accumulate.
>
> - Associative integration through AE-HMF, which recalls structural relations from semantically related base classes and uses them to guide cross-layer fusion when novel classes arrive.
>
> This leads to several distinctions compared with prior work:
>
> - **Different target challenge:**
>   FearNet [a] focuses on general incremental learning with replay, not the strict few-shot regime where covariance estimation becomes highly unstable. FoCAL [b] samples Gaussian pseudo-examples independently per class, without modeling cross-class covariance or multi-granularity feature structure.
>
> - **Different role of Gaussian modeling:**
>   Instead of replaying generated samples to fine-tune the models, we construct a multi-granularity Gaussian prior regularized by base-class covariance to guide layer-wise associative fusion,  helping maintain semantic alignment and preventing boundary drift, which earlier Gaussian approaches do not address.
>
> - **Different update mechanism:**
>   A lightweight fusion module is updated at incremental stages, trained on a small set of enhanced and real samples to dynamically adjust feature overlap and mitigate boundary drift. This preserves stability and reduces error accumulation—something replay-based or fully fine-tuned methods often struggle with.
>
> Empirically, **the effect of this reframed perspective is visible:** across CIFAR100 and miniImageNet, our method consistently achieves the lowest DR, while improving novel-class accuracy. These gains directly correspond to the two FSCIL failure modes our method is designed to mitigate.
>
> We have revised the paper to make these distinctions clearer and added explicit comparisons with FearNet and FoCAL. We sincerely thank the reviewer again for drawing attention to related work, which helped us better highlight the conceptual contribution of our approach.
>
> [a] FearNet: Brain-Inspired Model for Incremental Learning, ICLR 2018.
>
> [b] Ayub, A., & Fendley, C. (2022). Few-shot continual active learning by a robot. Advances in Neural Information Processing Systems, 35, 30612-30624.

---

> ### Author Response · Authors · 2025-11-24
> **Response to Reviewer xph2 (Part 2/4)**
>
> > **Q2: Complexity, hyperparameters, and computational/memory efficiency: is the complexity justified?**
>
> **A2.1:**
> Thank you very much for raising concerns about model complexity, computational/memory cost, and the potential hyperparameter burden. We appreciate this observation, and to address your concerns, we have added a full analysis of model complexity, efficiency, and hyperparameter sensitivity.
>
> - **Comparison of model complexity and computational efficiency**
>
> Our HDIN backbone is in the medium parameter range used in FSCIL works (Table 6 in the main paper). To ensure a fair comparison, we have maintained backbone sizes comparable to competing methods. Since we don't focus on designing lightweight network architectures, we compare with the similar concept method CoDF, whose performance is competitive with ours.
>
> Table 1: Model complexity and computational efficiency comparison with CoDF on CUB200.
>
> | Method | Training phase | Training Param | FLOPs  | Training Time | Inference Time（Average Sample） |
> | :----: | :------------: | :------------: | :----: | :-----------: | :------------------------------: |
> |  CoDF  | Cognitive task |  23.7M/23.7M   | 1.59G  |   233.91min   |             0.0171 s             |
> |  Our   |  First-phase   |  17.3M/17.3M   | 3.08 G |   26.64min    |             0.0953s              |
>
> From the table, we observe that the inference FLOPs and latency are indeed higher, which is mainly caused by the additional computations introduced to realise inter-layer interaction and associative modelling. However, our primary concern lies in the high cost of CoDF’s cognitive pretraining: its training time for the cognitive stage is nearly nine times longer than that of our single-stage training.
>
> - **Computational and memory costs in incremental phases.**
>
> Table 2: Parameter distribution across modules on CUB200.
>
> |            Module            | Parameter |
> | :--------------------------: | :-------: |
> |            Total             |   22.4M   |
> |             HDIN             |   17.1M   |
> |              fc              |   0.1M    |
> | feature_rectification.layer1 |   1.3M    |
> | feature_rectification.layer2 |   1.5M    |
> | feature_rectification.layer3 |   2.1M    |
> |       fc_layer.layer1        |   51.2K   |
> |       fc_layer.layer2        |   64.0K   |
> |       fc_layer.layer3        |   89.6K   |
> |       layer_attention        |   2.1K    |
>
> During incremental sessions, the HDIN backbone itself is frozen, we only fine-tune the `feature_rectification` and `layer_attention` modules, which minimizes computational overhead, GPU memory usage, and tuning burden. Additional memory costs primarily arise from the MVN distribution dictionary in AE-HMF (storing per-class and per-layer statistics) which scale linearly with the number of classes.  No rehearsal buffers, generative replay models are used. This significantly reduces tuning cost and prevents catastrophic forgetting.
>
> - **Hyperparameter analysis and tuning overhead**
>
> Although our method involves multiple loss weights, we provide stable default values for common datasets in both the main paper and the appendix (see the main paper and Appendix A.1). To address your concern, we analyze the key hyperparameter and the ablation results are added in Appendix A.4 in the revised PDF. Therefore, the practical tuning workload is much smaller than the reviewer might worry.
>
> Table 3: Sensitivity analysis of hyperparameter  $\lambda_{\text{cos}}$.
>
> | $\lambda_{\text{cos}}$ | 0.01  |  0.1  |  0.3  |    0.5    |  0.7  |  0.9  |
> | :--------------------: | :---: | :---: | :---: | :-------: | :---: | :---: |
> |          aACC          | 69.76 | 69.94 | 70.02 | **70.14** | 70.08 | 70.08 |
> |           DR           | 29.6  | 29.6  | 29.6  | **28.9**  | 29.6  | 29.5  |
>
> Table 4: Sensitivity analysis of hyperparameter $\lambda_{\text{nov}}$.
>
> | $\lambda_{\text{nov}}$ |  0.1  |    0.2    |  0.3  |  0.5  |  0.7  |  0.9  |
> | :--------------------: | :---: | :-------: | :---: | :---: | :---: | :---: |
> |          aACC          | 69.93 | **70.14** | 69.96 | 68.84 | 61.46 | 57.43 |
> |           DR           | 29.5  | **28.9**  | 29.5  | 30.0  | 31.9  | 38.2  |
>
> Table 5: Sensitivity analysis of hyperparameter $\lambda_{\text{glo}}$.
>
> | $\lambda_{\text{glo}}$ |  0.1  |  0.3  |    0.5    |  0.7  |  0.9  |
> | :--------------------: | :---: | :---: | :-------: | :---: | :---: |
> |          aACC          | 68.87 | 70.09 | **70.14** | 70.03 | 70.03 |
> |           DR           | 30.5  | 29.4  | **28.9**  | 29.6  | 29.6  |
>
> Table 6: Sensitivity analysis of hyperparameter $k$.
>
> | $k$  |   1   |     2     |   5   |  10   |
> | :--: | :---: | :-------: | :---: | :---: |
> | aACC | 69.83 | **70.14** | 70.12 | 70.08 |
> |  DR  | 30.1  | **28.9**  | 29.6  | 29.6  |

---

> ### Author Response · Authors · 2025-11-24
> **Response to Reviewer xph2 (Part 3/4)**
>
> > **Q2: Complexity, hyperparameters, and computational/memory efficiency: is the complexity justified?**
>
> **A2.2:**
> - **Is our method justified?**
>
> Thank you for asking whether the performance gain warrants the added components. We fully agree that architectural complexity must be justified by meaningful benefits. Through analysis of the model’s complexity, hyperparameters, and computational/memory costs, we demonstrate that the introduced components are justified, controllable, and do not pose an undue burden compared to baseline models. As for performance, CoDF reports results with ViT-T/16 and ViT-S/16; some of their best performances come from a larger backbone. When scaled to ViT-S, our method surpasses CoDF in both aACC and DR (Table 7), indicating that the proposed architectural idea is compatible with larger backbones and is capable of achieving SOTA.
>
> Table 7: Performance comparison with CoDF on CUB200 across different backbone architectures.
>
> |  Method  |          Backbone           |    0     |    1     |    2     |    3     | 4        |    5     |    6     |    7     |    8     |    9     |    10    |   aACC↑   |   DR↓   |
> | :------: | :-------------------------: | :------: | :------: | :------: | :------: | -------- | :------: | :------: | :------: | :------: | :------: | :------: | :-------: | :-----: |
> |   CoDF   |       ViT-T/16(5.6M)        |   82.5   |   79.6   |   76.8   |   72.2   | 72.3     |   69.5   |   69.0   |   68.2   |   67.1   |   67.1   |   67.0   |   71.88   |  19.4   |
> |   CoDF   |       ViT-S/16(21.6M)       |   87.6   |   85.3   |   83.5   |   80.5   | 80.5     |   78.0   |   77.7   |   77.7   |   76.8   |   76.8   |   76.3   |   80.05   |  12.6   |
> |   Ours   |   HDIN w/ ViT-T/16(17.3M)   |   84.5   |   81.8   |   79.9   |   78.0   | 76.5     |   74.6   |   74.2   |   74.3   |   73.0   |   73.5   |   73.0   |   76.66   |  13.6   |
> | **Ours** | **HDIN w/ ViT-S/16(33.8M)** | **87.3** | **85.7** | **85.1** | **82.9** | **81.7** | **80.3** | **80.1** | **80.1** | **79.5** | **79.6** | **79.4** | **81.97** | **9.0** |
>
> Our method achieves SOTA performance in terms of DR, demonstrating its effectiveness in mitigating catastrophic forgetting. The additional experiments confirm that higher absolute accuracy is attainable with stronger backbones, though our primary focus remains on developing cognitively-plausible learning mechanisms rather than purely pursuing SOTA numbers.
>
> We emphasize that our core contribution lies in exploring how machines can rapidly learn new concepts from a few examples—mirroring human cognitive processes. This involves:
>
> (1) Rapid, intuitive understanding of an object: recognizing which class it belongs to (common features) and what fine-grained details characterize it (specific features), which is modeled by our proposed HDIN network
>
> (2) Associative learning when encountering new concepts: humans naturally recall previously learned similar concepts, further facilitating the learning of new ones, which is modeled by our AE-HMF mechanism.
>
> We believe this cognitive-inspired approach offers a valuable perspective on addressing the stability-plasticity dilemma in FSCIL. Our quantitative and qualitative results validate that we have effectively modeled this process, and we continue to refine these ideas toward more elegant and efficient implementations in future work.

---

> ### Author Response · Authors · 2025-11-24
> **Response to Reviewer xph2 (Part 4/4)**
>
> > **Q3: The pretrained backbone setting in CUB-200 should be clarified in the main paper. The paper should also compare against simpler works that use pre-trained ImageNet features.**
>
> **A3:** Thank you for highlighting the need to clarify the pretrained-backbone setting on CUB-200 and for suggesting comparisons with methods that directly use ImageNet features. We agree that clarity on training protocols is important. You may have misunderstood our experimental setup. We strictly follow the standard protocol used in current mainstream FSCIL research: on CIFAR100 and miniImageNet, both our method and all comparison methods use feature backbones without any pretraining. On CUB-200, our method and all baselines consistently adopt pretraining backbones. While a few recent works have explored using large pretrained backbones for FSCIL across three benchmarks, the mainstream and widely accepted protocol does not employ additional pretraining, and we follow this standard setting to ensure fair and direct comparison. Under this fair protocol, our method achieves the lowest DR scores on all benchmark datasets, which clearly demonstrates the advantages of the proposed approach. We have already clarified the backbone setup explicitly in the main paper.
>
>
>
> > **Q4: What variance schedule, class-adaptive covariance, and safeguards are used in  Gaussian sampling to prevent prototype collapse?**
>
> **A4:** Thank you for raising this technical concern—Gaussian instability is indeed a known risk in FSCIL. We agree that our main paper could explain these design choices more clearly.  The key mechanisms: **Covariance is smoothed through average scheduling** and **multi-granularity Gaussian priors** reduce harmful over-smoothing; **Joint training with mixed samples** and **the real sample prototype rewrite policy** prevent prototype collapse.
>
> - **Variance scheduling** **and smoothing**
>
> In sampling, instead of directly using a single covariance, we compute the mean of covariances of the k most-similar base classes to reduce the chance of sampling from semantically distant distributions that would cause class drift. Further, we add a small diagonal jitter (eps * I) before forming the multivariate normal to smooth the sharp distribution of fine-grained categories and alleviate numerical problems.
>
> - **Class-adaptive covariance**
>
> We construct covariances for the Final feature and each Layer feature separately, resulting in multi-granularity Gaussian priors for each class. The covariance of each layer is derived from the historical class statistics of the corresponding layer, thus maintaining the consistency of the intra-layer structure.
>
> - **Safeguards against prototype collapse**
>
> Multi-source loss constraint: enhanced samples and real support samples are jointly trained; after layer fusion, we apply both cosine alignment and cross-entropy objectives. This prevents optimization from relying solely on synthetic prototypes.
>
> Prototype rewrite policy: final prototypes are always updated from the mean of real samples in the base/novel sessions. Enhanced samples are used in the incremental training and never overwrite the prototype obtained from the basic session.
>
> For your advice, we have already added more details about IGS in the revised paper and provided a pseudocode about the mechanisms in Appendix B.

---

### Official Review · Reviewer_tWxW · 2025-11-03

**Soundness:** 3
**Presentation:** 3
**Contribution:** 3
**Rating:** 6
**Confidence:** 3

**Summary:**

This paper investigates the challenging problem of Few-Shot Class-Incremental Learning (FSCIL), where a model must incrementally learn new classes from few examples while retaining previously acquired knowledge. The authors propose a cognition-inspired dual-module framework comprising a Hierarchical Dual-Stream Interaction Network (HDIN) that fuses local (ResNet-based) and global (ViT-based) representations through channel-adaptive attention, and an Associative-Enhanced Hierarchical Memory Fusion (AE-HMF) module that utilizes Gaussian sampling of class prototypes for cross-layer associative memory consolidation. Experiments on CIFAR100, miniImageNet, and CUB200 benchmark datasets demonstrate their method’s effectiveness in achieving lower performance degradation rates and competitive accuracy, without relying on large-scale pretraining or data augmentation.

**Strengths:**

1. Intuitive approach to deal with the FSCIL problem.
2. Strong performance improvement in terms of the lowest performance drop for CIFAR100 and miniImageNet datasets

**Weaknesses:**

1. Poor performance on the CUB dataset. The authors should discuss the reason behind this clear gap from the best method. Is it because of larger image size or a finegrained setup, etc.
2. The CoDF method seems to perform better in terms of accuracy, even though, as the authors mention, it requires more epochs for training. However, the authors should describe in more detail why the proposed approach should be preferred over CoDF apart from the training epochs.

**Questions:**

1. What is the reason behind the clear gap from the best method on the CUB dataset. Is it because of larger image size or a finegrained setup, etc.
2. Why the proposed approach should be preferred over CoDF apart from the training epochs.?
3. Does the proposed approach use any additional information that CoDF doesnot use?

---

> ### Author Response · Authors · 2025-11-24
> **Response to Reviewer tWxW (Part 1/2)**
>
> We sincerely thank you for the careful reading and for raising important concerns regarding the CUB-200 performance, the comparison with CoDF, and the use of additional information. Following your suggestions, we have expanded our analyses, added new experiments, and revised the explanations accordingly. Below we address each point in detail.
>
> > **W1&Q1:  Why our method lags behind the best on CUB-200?** **Is it because of larger image size or a finegrained setup, etc.**
>
> **A1:** The gap is primarily due to the fine-grained of CUB-200, CoDF’s choice of backbone, and several training tricks. This not because we used a different image resolution or a special data split. The backbone-size differences mainly caused this gap.
>
> - **The characteristics of the dataset:** CUB-200 is a fine-grained dataset in which different classes share many visual similarities. The MAE backbone used in CoDF can produce richer representations of such subtle visual patterns. As reported in [1], CUB-200 belongs to a category of scenarios in which CoDF is particularly effective, performing even better than on miniImageNet and CUB-200. However, our method focuses more on disentangling common and specific patterns across tasks, and is therefore less sensitive than CoDF to fine-grained differences between categories.
> - **The optimization of training strategy:** CoDF introduces an Extended Warm-Up (EWU) strategy, bringing a considerable performance improvement in the reported ablation study [1]. Our method does not use any sophisticated training tricks of this kind.
> - **The backbone configuration:** To ensure that model sizes are on a comparable basis, the ViT blocks used in the common stream of HDIN follow the same configuration as ViT-T. CoDF reports results with ViT-T/16 and ViT-S/16; some of their best performances come from a larger backbone.
>
> We agree that this may have led to an underestimation of our full potential in the initial submission. To address this, we have added a new experiment where HDIN is equipped with ViT-S blocks. As shown in Table 1 (now also added to the paper), our method’s performance improves and surpasses the best reported results, which indicates that **the proposed architectural idea is compatible with larger backbones and can reach SOTA under that setting**.
>
> Table 1: Performance comparison with CoDF on CUB200 across different backbone architectures.
>
> |  Method  |          Backbone           |    0     |    1     |    2     |    3     |    4     |    5     |    6     |    7     |    8     |    9     |    10    |   aACC↑   |   DR↓   |
> | :------: | :-------------------------: | :------: | :------: | :------: | :------: | :------: | :------: | :------: | :------: | :------: | :------: | :------: | :-------: | :-----: |
> |   CoDF   |       ViT-T/16(5.6M)        |   82.5   |   79.6   |   76.8   |   72.2   |   72.3   |   69.5   |   69.0   |   68.2   |   67.1   |   67.1   |   67.0   |   71.88   |  19.4   |
> |   CoDF   |       ViT-S/16(21.6M)       |   87.6   |   85.3   |   83.5   |   80.5   |   80.5   |   78.0   |   77.7   |   77.7   |   76.8   |   76.8   |   76.3   |   80.05   |  12.6   |
> |   Ours   |   HDIN w/ ViT-T/16(17.3M)   |   84.5   |   81.8   |   79.9   |   78.0   |   76.5   |   74.6   |   74.2   |   74.3   |   73.0   |   73.5   |   73.0   |   76.66   |  13.6   |
> | **Ours** | **HDIN w/ ViT-S/16(33.8M)** | **87.3** | **85.7** | **85.1** | **82.9** | **81.7** | **80.3** | **80.1** | **80.1** | **79.5** | **79.6** | **79.4** | **81.97** | **9.0** |
>
> [1] Xuan Wang, Zhong Ji, Yanwei Pang, and Yunlong Yu. A cognition-driven framework for few-shot class-incremental learning. *Neurocomputing*, 600:128118, 2024b.

---

> ### Author Response · Authors · 2025-11-24
> **Response to Reviewer tWxW (Part 2/2)**
>
> > **W2&Q2:  Why prefer our approach over CoDF beyond training epochs?**
>
> **A2:** We apologize for not providing a more thorough comparison between our method and CoDF to better highlight the advantages of our work. We clarify the following advantages of our method:
>
> - **No costly self-supervised pretraining.** CoDF’s cognition task pretraining improves representations but requires substantial pretraining cost caused by the MAE backbone. Our method achieves effective feature disentanglement without such a heavy training cost.
> - **Structural decoupling.** CoDF separates cognitive roles via task-differentiated optimization. We explicitly disentangle common and specific features in the network architecture, which is more flexible in incremental scenarios where one cannot rely on pretraining every time new datasets come.
> - **Explicit adjustment of novel-class decision boundaries by the AE-HMF module that improves base/novel-class separability and yields a more useful base/novel trade-off.** Under the FSCIL task setting, base classes account for about 50%–60% of all categories, so the overall performance is mainly driven by base-class performance. CoDF freezes the backbone after base training and builds novel classifiers with few supports, which can lead to overlapping decision boundaries in the feature space defined by base classes. Our AE-HMF module actively adapts the decision boundaries for novel classes (using cross-layer feature fusion trained by enhanced and real samples), improving the separability of novel classes at incremental stages. As shown in the table below and in Figure 4 of the paper, the performance on base classes slightly decreases, while that on novel classes improves significantly, leading to a more favorable trade-off. We believe this phenomenon is of greater practical value for FSCIL scenarios, where the model should genuinely learn the novel classes while still maintaining the performance on base classes.
>
> Table 2: AE-HMF ablation: Impact on Novel Classes.
>
> |     Method     |     1-100      |     101-110     |    111-120    |    121-130     |    131-140     |    141-150     |     151-160     |    161-170     |    171-180     |    181-190     |    191-200     | Final acc↑ |
> | :------------: | :------------: | :-------------: | :-----------: | :------------: | :------------: | :------------: | :-------------: | :------------: | :------------: | :------------: | :------------: | :--------: |
> |   w/ AE-HMF    |      86.7      |      59.9       |     68.3      |      59.4      |      79.7      |      53.4      |      63.2       |      72.1      |      72.9      |      78.5      |      82.3      |    77.7    |
> | **w/o AE-HMF** | **85.9(-0.8)** | **72.0(+12.1)** | **72.1(3.8)** | **60.1(+0.7)** | **81.3(+1.6)** | **57.0(+4.4)** | **73.3(+10.1)** | **73.7(+1.6)** | **75.6(+2.7)** | **83.0(+4.5)** | **82.0(-0.3)** |  **79.4**  |
>
>
>
> > **Q3:  Do we use additional information that CoDF does not?**
>
> **A3:** We appreciate the reviewer’s careful attention to potential information usage. We clarify in the revised manuscript that we do not use any additional labeled data, external tasks, or side information beyond what CoDF uses. What we leverage is the statistical information of base-class prototypes (which structure-based FSCIL methods already store) more effectively for training the AE-HMF module. As illustrated in Fig. 3(b), when adjusting the novel classes, we leverage the statistical information of class prototypes stored during base-class training to perform Gaussian sampling, which supports the training of the feature fusion module in the novel-class stage. We just make more effective use of this already stored information to facilitate learning of the novel classes and preserving the old classes.

---

### Author Response · Authors · 2025-11-30
**General Response**

We sincerely thank Reviewers tWxW, xph2, and uqHE for their careful reading and constructive suggestions.  In the following, we summarize the main improvements made in the revision (all changes in the main manuscript are marked in blue), and we remain fully available for any additional inquiries.

**Main manuscript:**

- **Lines 363–367:** Clarified the experimental protocol for CUB-200, including pretraining settings for fair comparison.
- **Lines 132–135:** Added clearer conceptual distinctions from prior work to highlight novelty.
- **Lines 332–346:** Expanded descriptions of covariance construction, top-k selection, sampling strategy, training pipeline, and prototype update rules.
- **Lines 405–410:** Expanded comparison between a cognition-driven method CoDF beyond training cost.
- **Figure 5:** Moved the key motivational figure from the supplement into the main paper and improved notation consistency.
- **Figures 1–3:** Redrawn with explicit data-flow arrows and unified notation to improve readability.

**Appendix:**

- **Appendix A.2:** Added aligned ViT-S scalability experiments demonstrating that HDIN surpasses CoDF on CUB-200 under matched backbones.
- **Appendix A.4:** Provided additional ablations illustrating robustness of the key hyperparameters and module contributions.

- **Appendix B:** Added pseudocode for IGS to clarify the training and update rules.

Based on their feedback we grouped all concerns into six common categories and addressed each with **targeted modifications**, **new experiments**, **detailed implementation descriptions**, and **clarity improvements**.

> **Dataset / CUB-200 Performance Gap.  (@tWxW, xph2, uqHE)**

- We added aligned experiments with a stronger backbone (HDIN + ViT-S) and HDIN **surpasses CoDF on CUB200** (higher aACC, lower DR), showing the observed gap is due to backbone sensitivity and the fine-grained nature of CUB 200, not a fundamental issue with our approach. (See Table 4, Appendix A.2.)

> **More Comparison with CoDF. (@tWxW, uqHE)**

- We emphasize methodological advantages: HDIN **explicitly disentangles common vs. specific representations**, and AE-HMF **actively adjusts the novel–base decision boundary** during incremental stages via multi-layer fusion and enhanced samples — improving the base/novel trade-off in practice.
- We provide a detailed efficiency comparison; overall training time is significantly shorter because CoDF requires expensive cognitive pretraining. During the incremental stage, our backbone is frozen and only lightweight modules are fine-tuned, keeping incremental compute and memory costs low.

> **Stability of Gaussian Sampling & Prototype Collapse. (@xph2, uqHE)**

- We added implementation details and safeguards:
  - Top-k covariance averaging with diagonal jitter for numerical stability and smoothing.
  - Multi-granularity covariance construction rather than a single global covariance.
  - Training with mixed samples: enhanced samples are trained together with real supports under joint losses.
  - Prototype protection policy: final prototypes are always rewritten using real samples only.
- We included sensitivity studies for the key hyperparameter $k$ in Appendix A.4, showing stable performance within reasonable parameter ranges.

> **Complexity / Module Count / Computational and Memory Efficiency. (@xph2, uqHE)**

- We added a comprehensive complexity table comparing parameters, FLOPs, training time, and inference latency against CoDF.
- We emphasize practical trade-offs: during incremental sessions the backbone is frozen and only small modules are updated, which minimizes training cost and GPU memory usage.
- We included ablation and sensitivity analyses demonstrating that most hyperparameters are robust and that reasonable default settings work well (Appendix A.4).

> **Novelty and Distinction from Prior Work. (@xph2)**

- We reframed the contribution: our work is a **systematic solution targeted at two underexplored failure modes** in FSCIL—**Intra-class variance collapse**, and **Decision-boundary drift** caused by extremely limited novel supports.
- We added explicit related discussion in the main text (L132–135). And We clarified differences from related methods (FearNet, FoCAL):
  - **HDIN**: dual-stream, multi-granularity, cross-layer interactions that prevent variance collapse rather than fusing features only at output level.
  - **AE-HMF**: top-k guided, multi-layer covariance priors that regularize Gaussian sampling and drive associative cross-layer fusion — i.e., priors are used for stabilizing cross-layer fusion, not naive replay.

> **Readability need to be improved. (@uqHE)**

- We redrew Figures 1–3 with explicit data-flow arrows and rewrote ambiguous text passages for clarity.
- We added more detailed descriptions for covariance construction, top-k selection, sampling, and prototype update rules in the main text (L332–346) and a corresponding pseudocode in Appendix B.

---

### Author Response · Authors · 2025-11-30
**Summary of the Rebuttal Period**

We would like to express our sincere gratitude to the area chair and all reviewers for the time and effort you have invested during the review and rebuttal phases. Given the limited time before the discussion period concluded, we provide here a concise summary of the key points regarding our manuscript.

In this work, we present **a cognition-driven framework designed to address two fundamental failure modes in FSCIL: intra-class variance collapse and decision-boundary drift under few-shot incremental learning**. Our approach consists of HDIN, which disentangles and integrates multi-granularity local–global cues for stable representation learning, and AE-HMF, which performs information-guided Gaussian associative fusion to adapt class boundaries under few-shot conditions. This design directly targets the root causes of forgetting and imbalance in FSCIL, rather than relying on heavier pretraining or heuristic regularization. Comprehensive experiments on CIFAR100, miniImageNet, and CUB-200, along with ablations, sensitivity studies, and complexity analyses, demonstrate the effectiveness and robustness of our paradigm.

In the initial reviews, **the reviewers acknowledged the motivation behind our formulation, the soundness of the methodological design, and the empirical gains on CIFAR100 and miniImageNet**. Several reviewers also highlighted the clear presentation and the relevance of our contributions to addressing practical FSCIL challenges.

**Reviewer tWxW** focused on our performance on CUB-200, the substantive advantages over CoDF beyond training time, and whether any additional information was used. We addressed these concerns by adding fair-capacity comparisons, clarifying the architectural and learning benefits of our approach, and confirming that no extra data or labels were used.

**Reviewer xph2** was mainly concerned with the novelty relative to prior cognition-inspired and Gaussian-based methods, the model’s complexity and hyperparameter cost, clarity of the pretraining setup, and the stability of Gaussian sampling. We responded by clarifying the unique contributions of our framework, providing added efficiency and sensitivity analyses, making the experimental setup explicit, and explaining the sampling mechanism and stabilization strategies.

**Reviewer uqHE** emphasized the clarity of figures and module descriptions, details of the Gaussian sampling process and parameter updates, the robustness of top-k selection, and quantitative efficiency comparisons with CoDF. We addressed these concerns by redrawing figures, adding pseudocode and explanations, including sensitivity experiments, and supplementing training-/inference-time and resource-usage comparisons.

These additions materially strengthened the manuscript’s completeness and rigor. Up to the point when the discussion was interrupted, one reviewer (uqHE) responded positively to our rebuttal, **indicated that the major concerns had been addressed, and explicitly updated their score accordingly.** This gives us cautious optimism that our clarifications may similarly resolve the shared concerns of other reviewers.

We sincerely hope that the revisions and additional analyses we provided satisfactorily address all reviewer concerns. We are deeply grateful for the thoughtful feedback and constructive guidance throughout this process. Thank you once again for your careful and fair evaluation of our work.

---

### Meta-Review · Area_Chair_jRcm · 2026-01-03

**Summary:**

This paper focuses on few-shot class-incremental learning using a cognition-inspired approach. Three experts participated in the review process, and the paper received mixed ratings (one positive and two negative). The main initial concerns can be summarized as follows: (i) inferior performance on the CUB-200 dataset compared to state-of-the-art methods [tWxW]; (ii) limited novelty, as existing works already incorporate cognition-inspired and prototype-based approaches [xph2]; (iii) excessive method complexity, with a large number of modules and hyperparameters but only marginal performance gains [xph2, uqHE]; and (iv) unclear effectiveness of the proposed Gaussian sampling strategy [xph2, uqHE].

The AC carefully reviewed the paper, the reviews, and the author responses. I agree with [xph2, uqHE] that the proposed method is overly complex, involving many modules and hyperparameters, while delivering only marginal performance improvements, particularly in comparison with CoDF. Although the authors provide additional experiments using backbones consistent with CoDF and report improved results, these findings do not convincingly demonstrate that the gains primarily stem from the proposed motivations and corresponding design choices.

First, the proposed method integrates two pretrained backbones into the system, which already introduces substantial prior knowledge compared to other state-of-the-art methods. As shown in Table 3, using only LSS yields an aAcc of 61.7, while combining LSS with GGS increases the aAcc to 68.9, representing a gain of 7.2 points. This performance is already among the highest reported in Table 1, excluding CoDF. Furthermore, Table 3 shows that the addition of three losses (cos, novel, and global), the replay mechanism, and the associated hyperparameters contributes only an additional 1.2 points in aAcc. Moreover, according to the authors’ discussion of hyperparameter sensitivity, these additional components may even have negative effects. A similar observation can be drawn from the authors’ response indicating that the “w/ AE-HMF” configuration already outperforms CoDF. This raises a key question: is the overall improvement primarily due to the introduction of an additional network that provides a stronger baseline, or is it attributable to the proposed method itself? While the authors may argue that the LSS + GGS configuration is part of their design, the stand-alone contributions of the additional model components and the specific design choices are not sufficiently justified.

From the AC’s perspective, several issues require further clarification:

1.	Motivation and empirical validation:
The authors motivate their approach by claiming that collapsed intra-class variance and boundary instability are observed in prior works. However, no empirical validation or experimental evidence is provided to demonstrate that existing methods indeed suffer from these limitations. Although the t-SNE visualization in Figure 5 shows related patterns, it is unclear which specific components of the proposed method lead to these observations. For a clearer storyline, the authors are encouraged to first present evidence of these issues in prior works before introducing the proposed method. Simply explaining these claims textually is not sufficiently convincing.

2.	Claims about CNN and ViT properties:
The discussion in Lines 209–214 regarding the properties of CNNs and ViTs is not fully justified. Relevant references or empirical validation should be provided to support these claims.

3.	Loss formulation and potential degeneracy:
Starting from Line 273, the loss function in Equation (7) lacks sufficient detail. It is unclear how positive and negative pairs are constructed. Moreover, Equation (7) aims to improve similarity in the first term involving f_fused and f_final. However, since f_fused already contains f_final as defined in Equation (5), the network could trivially learn to ignore MLP_final and the first part of MLP_fused, resulting in f_fused effectively collapsing to f_final. It is unclear how this potential degeneracy is addressed.


Overall, given the above observations, the AC believes that substantial revisions are required, and the paper, in its current form, is not ready for publication at ICLR.

**Reviewer Concerns:**

The concerns regarding (ii) and (iv) have been adequately addressed. However, the major issues related to inferior performance, limited novelty, and an overly complex solution with only marginal improvements remain unresolved.

**Reviewer Scores:**

Reviewer [tWxW] may remain positive, while [uqHE] may or may not increase the score, as the method complexity is still not sufficiently justified. Reviewer [xph2] is expected to remain negative, since the major concerns have not been addressed.

---

### Decision · Program_Chairs · 2026-01-26

Reject